# Unravelling the developmental and functional significance of an ancient Argonaute duplication

Arie Fridrich [1,3], Vengamanaidu Modepalli [1,2,3], Magda Lewandowska [1], Reuven Aharoni[1] & Yehu Moran [1✉]

MicroRNAs (miRNAs) base-pair to messenger RNA targets and guide Argonaute proteins to mediate their silencing. This target regulation is considered crucial for animal physiology and development. However, this notion is based exclusively on studies in bilaterians, which comprise almost all lab model animals. To fill this phylogenetic gap, we characterize the functions of two Argonaute paralogs in the sea anemone *Nematostella vectensis* of the phylum Cnidaria, which is separated from bilaterians by ~600 million years. Using genetic manipulations, Argonaute-immunoprecipitations and high-throughput sequencing, we provide experimental evidence for the developmental importance of miRNAs in a non-bilaterian animal. Additionally, we uncover unexpected differential distribution of distinct miRNAs between the two Argonautes and the ability of one of them to load additional types of small RNAs. This enables us to postulate a novel model for evolution of miRNA precursors in sea anemones and their relatives, revealing alternative trajectories for metazoan miRNA evolution.

[1] Department of Ecology, Evolution and Behavior, Alexander Silberman Institute of Life Sciences, Faculty of Science, The Hebrew University of Jerusalem, 9190401 Jerusalem, Israel. [2] The Marine Biological Association of the United Kingdom, Citadel Hill, Plymouth, UK. [3]These authors contributed equally: Arie Fridrich, Vengamanaidu Modepalli. ✉email: yehu.moran@mail.huji.ac.il

microRNAs (miRNAs) are pivotal players in post-transcriptional gene regulation in animals and plants. As such, they carry important roles in many developmental and physiological processes in both kingdoms[1–4]. Whereas their mode of action and functional roles are extensively studied in bilaterian animals such as vertebrates, nematodes and arthropods, they are understudied in non-bilaterian animals such as cnidarians and sponges despite being identified in these animal phyla more than a decade ago[5]. Until now, there is only little experimental evidence supporting any developmental or physiological role for miRNAs in organisms other than bilaterian animals and land plants, despite their wide phyletic distribution[6,7]. Our previous studies revealed that the miRNAs of Cnidaria (sea anemones, corals, jellyfish, and hydroids), the sister group of Bilateria, exhibit remarkable similarities to plant miRNA biogenesis and mode of action[8–10]. This was unexpected, as many contemporary studies argue that the vast differences between the plant and animal miRNA systems point towards convergent evolution and lack of miRNAs in the last common ancestor of animals and plants[6,11,12]. Among the similarities found between plant and cnidarian miRNAs, arguably the most striking one is the frequent tendency of the miRNA-guided RNA-induced Silencing Complex (RISC) of cnidarians to cleave ('slice') its targets[9]. The key proteins of RISC that perform the slicing in all miRNA-bearing organisms are the Argonautes (AGOs)[13–15]. Their ability to slice is probably an ancient property of AGOs that pre-existed before the appearance of miRNA pathways in plants and animals[16,17]. Bioinformatic analysis revealed that members of the Hexacorallia cnidarian subclass (sea anemones and reef-building corals) carry two AGOs with intact RNAse catalytic sites[10]. As hexacorallians diverged roughly 540 million years ago (MYA)[18,19], this gene duplication seems to be extremely stable and suggests that each of the two AGOs specialized in a different role[10].

To test the role of miRNAs in Cnidaria and the hypothesis of AGO specialization in Hexacorallia, in our current work we took advantage of genetic-manipulation tools available for the sea anemone Nematostella vectensis as well as immunoprecipitation (IP) and high-throughput sequencing techniques. Our results provide direct evidence that indeed miRNAs play a significant role in the development of a cnidarian and reveal the specialization of the two AGO proteins of Hexacorallia. Moreover, the results also enable us to postulate a new hypothesis regarding how novel miRNAs are born and further evolve in Hexacorallia.

## Results

**Nematostella development requires two AGO paralogs that duplicated >500 MYA.** To get a broader insight into the evolutionary history and fate of AGO duplications in Metazoa with a focus on Cnidaria, we constructed AGO phylogeny in which we included bilaterian as well as cnidarian representatives (Fig. 1a). This phylogeny revealed that while the first Nematostella AGO, NveAGO1, clusters with AGO1 proteins of other hexacorallians, the second Nematostella AGO, NveAGO2, is positioned in a separate clade, together with the hexacorallian AGO2. Interestingly, we could detect at least one AGO1 ortholog and one AGO2 ortholog in most hexacorallian transcriptomes or genomes we surveyed. Our phylogeny suggests that the two AGO paralogs duplicated before the split of sea anemones and reef-building corals and co-existed for more than 500 million years. Such long co-existence of paralogs strongly hints towards sub- or neo-functionalization, otherwise we would expect a frequent loss of one of the paralogs over hundreds of millions of years. Our previous results suggested that destabilization of miRNAs and inhibition of their biogenesis by knocking down the

methyltransferase NveHEN1 or the RNAse III NveDicer1, respectively, results in severe defects in Nematostella early development[8]. However, these components also play a role in the stabilization and biogenesis of other small RNAs (sRNAs) and may be involved in additional pathways[6,8]. Thus, we decided to test the potential specialization of the Nematostella AGOs and the role of miRNAs in Nematostella development by inhibiting the expression of each AGO separately. To this end, we microinjected Morpholino Antisense Oligos (MOs) against NveAGO1 and NveAGO2 to Nematostella zygotes (Two different translation inhibition MOs for each NveAGO). AGO morphants, but not zygotes injected with a control MO, failed to reach the primary polyp stage nine days post fertilization (dpf) (Fig. 1b, c, Supplementary Fig. 1). These morphological phenotypes are grossly similar to the phenotypes that we previously obtained when we inhibited other components involved in the miRNA pathway[8]. The specificity of each of the knockdowns (KD) was assayed by Western blot with specific antibodies raised against NveAGO1 and NveAGO2 that were generated for this study (see "Methods"). The bands that correspond to the expected size of NveAGO1 and NveAGO2 are absent or strongly downregulated in morphants (Fig. 1d, Supplementary Fig. 1b). The specificity of the antibodies was further verified by immunoprecipitation (IP) performed on lysates from primary polyps (9-days old) followed by liquid chromatography coupled to tandem mass spectrometry (LC-MS-MS) (Fig. 1e, Supplementary Data 1). The LC-MS/MS analysis revealed that in NveAGO1 IP, NveAGO1 peptides (but not NveAGO2) were enriched compared to the control sample incubated with Rabbit IgG. Similarly, NveAGO2 IP samples were enriched for NveAGO2 (but not NveAGO1) peptides (Fig. 1e, Supplementary Data 1). The differences were statistically significant ($P \leq 0.001$) and these results support the specificity of the custom antibodies.

While the failure in metamorphosis of the morphants of each AGO KD is an indication for functional specialization of each paralog[20], we could not exclude the possibility that the AGOs are redundant in their function but their high dosage is critical for development[21]. Thus, we performed RNA-seq of three days old control and morphant animals and looked at the differences in their transcriptomic signatures (Fig. 2, Supplementary Fig. 2, Supplementary Data 2). While the controls of all experiments clustered together, NveAGO1 and NveAGO2 morphants greatly differed from one another, pointing towards specialization rather than redundancy and dosage-dependency. Since the main biological function of AGOs in animals is to guide sRNAs to target other RNA transcripts[22,23], we hypothesized that these transcriptomic differences might stem from different sRNA populations that are carried by these two AGOs, affecting different sets of RNA targets. Notably, hexacorallian AGO1 forms a sister group to the known bilaterian miRNA-AGOs. Contrastingly, the hexacorallian AGO2 is positioned outside these two sister groups (Fig. 1a), leading us and others to predict it will not bind miRNAs but another group of sRNAs[24]. However, because phylogeny alone is insufficient to determine the differences between the NveAGO1 and NveAGO2 sRNA cargos, we continued with an experimental approach to test that.

**AGO IP uncovers novel Nematostella miRNAs.** First, we confirmed by western blot that the antibodies αNveAGO1 and αNveAGO2 are suitable for a specific IP in three distinct developmental stages (planula, primary polyp and adult) (Fig. 3a). For each developmental stage, we immunoprecipitated each NveAGO in two biological replicates, and included two replicates of rabbit IgG IP as negative controls. Next, we generated sRNA libraries from IP-extracted, size-selected RNA (see "Methods") using a

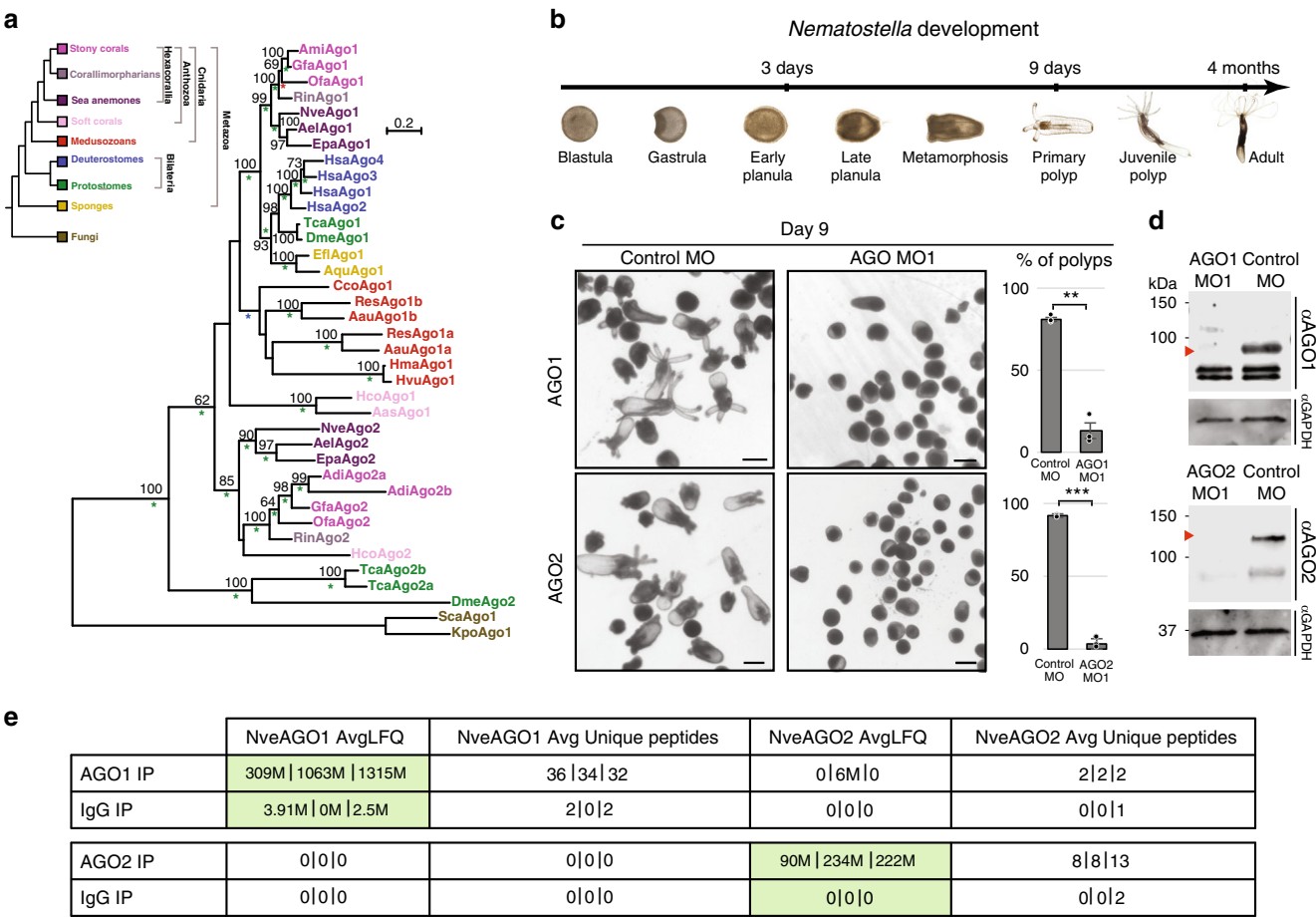

**Fig. 1 The hexacorallian AGO duplication, and its importance for *Nematostella* development. a** A phylogenetic relationship of metazoan AGOs. The tree was constructed with the LG model (+I, +G). Bootstrap support values above 50% are indicated above branches. Posterior probability values of a Bayesian tree of the same topology are indicated by asterisks. A green asterisk represents a value of 1.0, blue represents values >0.95 and lower than 1.0 and a value of 0.95 is indicated by a red asterisk. The tree was rooted by the AGOs of the fungi *Saccharomyces castellii* (Sca) and *Kluyveromyces polysporus* (Kpo). Abbreviations of other species names are: Aas, *Acanthogorgia aspera*; Aau, *Aurelia aurita*; Ael, *Anthopleura elegantissima*; Adi, *Acropora digitifera*; Ami, *Acropora millepora*; Aqu, *Amphimedon queenslandica*; Cco, *Craterolophus convolvulus*; Dme, *Drosophila melanogaster*; Efl, *Ephydatia fluviatilis*; Epa, *Exaiptasia pallida*; Gfa, *Galaxea fascicularis*; Hco, *Heliopora coerulea*; Hma, *Hydra magnipapillata*; Hsa, *Homo sapiens*; Hvu, *Hydra vulgaris*; Nve, *Nematostella vectensis*; Ofa, *Orbicella faveolata*; Res, *Rhopilema esculentum*; Rin, *Rhodactis indosinensis*; Tca, *Tribolium castaneum*. Protein sequences are available in Supplementary Data 10. **b** Timeline of *Nematostella* development. **c** Animals injected with control MO (left panels) 9 dpf. **c** Animals injected with NveAGO1 and NveAGO2 MOs (middle upper and lower panels, respectively) 9 dpf. Majority of NveAGO1 and NveAGO2 depleted animals did not reach primary polyp stage (right upper and lower panels, respectively) at 9 dpf, three independent biological replicates. **P = 0.0069, ***P = 0.0005 (one tailed Students *t*-test), (Supplementary data 6). Data are presented as mean values +/− SD. Scale bars are 250 μm. **d** Western blot validation of knockdowns with NveAGO1 and NveAGO2 custom antibodies on extracts from 3 days old planulae, each blotting experiment was carried out twice independently. **e** LC-MS/MS analysis on three technical replicates. In NveAGO1 IP, an average of 34 specific peptides corresponding to NveAGO1, yielding an average label-free quantification (LFQ) value of ~895 million, whereas the control sample incubated with Rabbit IgG rather than αNveAGO1 yielded 1.3 peptides and an average LFQ value of 2.1 million. NveAGO2 IP samples provided an average of 9.6 peptides specific for NveAGO2 and an average LFQ value of ~182 million vs. 0.6 peptides and LFQ of 0 for the IgG control. Green shaded pairs indicate statistical significance (P = 0.0003 for NveAGO1, and P = 0.001 for NveAGO2, Statistical analysis was performed using the Perseus statistical package, see "Methods").

modified protocol suitable for lower sRNA quantities[25]. mir-Deep2[26] was used for de novo identification and quantification of novel and known *Nematostella* miRNAs. To minimize false positives, we used stringent criteria, which were previously suggested for miRNA annotation in bilaterians[27]. Specifically, a novel bona fide miRNA was considered if it exhibits: (a) >3-fold enrichment in the IP sample compared to the negative controls, (b) a clear signature for strand selection, with a dominant guide strand, that contains a homogeneous 5′ end, (c) guide/star ratio higher than two, (d) presence in both biological replicates with a minimum of 70 reads from the guide strand in at list one replicate, (e) a minimum of 16 hybridized nucleotides in the predicted guide/star duplex (see example in Fig. 3b, and Supplementary

Data 3). Since strand selection in *Nematostella* is significantly stronger on average than in bilaterians[5], starless miRNAs that passed the described criteria were also considered, however, classified as a separate category (Supplementary Data 4). In all libraries, the fraction of reads that mapped to miRNAs was higher than in the IgG-IP negative controls (Fig. 3c, Supplementary Data 4). In addition, we assessed the enrichment for miRNAs in sRNA libraries that were constructed from whole animals in the current study ("Input"; Supplementary Data 4, see "Methods") as well as in an independent data from previous study[9] (Supplementary Data 4). Altogether, we identified 86 novel miRNAs (of which 19 were classified as "starless") (Fig. 3b, Supplementary Data 4), increasing the total number to 138, with 42 previously

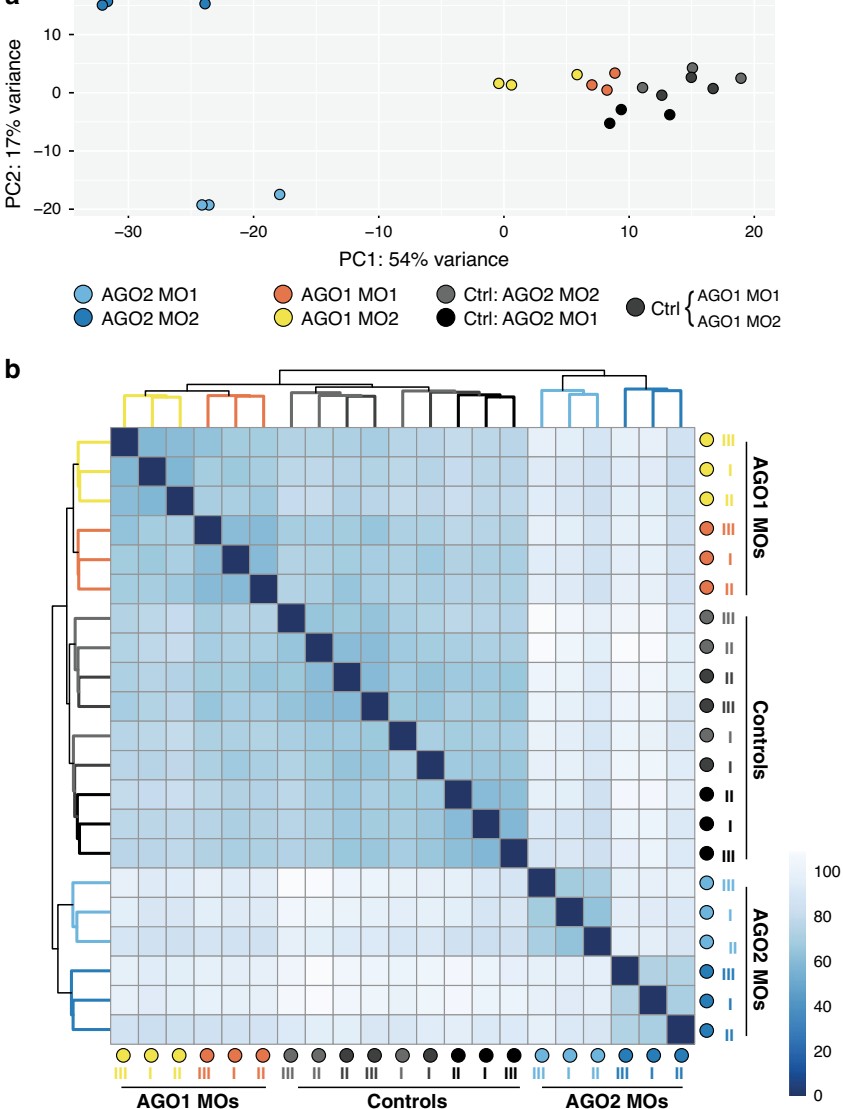

**Fig. 2 NveAGO1 and NveAGO2 knockdowns exhibit distinct transcriptomic signatures. a** Principal component analysis (PCA) plot and dendrogram (**b**) exhibiting differences in transcriptomic signatures for three distinct biological replicates of NveAGO1 and NveAGO2 knockdowns (two different MOs for each NveAGO). NveAGO1 MO1 and NveAGO1 MO2 were injected in parallel to the same control MO. STAR[66] was used to align reads to *Nematostella* genome.

identified miRNAs being revoked for not passing the precursor structure and homogeneity criteria, or for not being enriched in AGO IP compared to the IgG IP. Of the 52 previously identified remaining miRNAs, 49 overlapped with the list of Moran et al.[9] (which contained 87 miRNAs) and 31 with the list of Grimson et al.[5] (which contained 40 miRNAs) (see the updated list of miRNAs in Supplementary Data 4).

The reason so many miRNAs have not been identified earlier, stems from the fact that in previous studies, sRNAs were collected from whole animals. In *Nematostella*, unlike most bilaterians, such libraries are predominated by another class of sRNAs: the piRNAs[5,9]. Unlike most bilaterians, *Nematostella* piRNAs are not restricted to the germline, but are rather expressed broadly in somatic tissues[28]. Hence, NveAGO IP enabled us to overcome this limitation and reliably identify a significantly larger set of lower expressed novel miRNAs.

**NveAGO1 and NveAGO2 exhibit differential miRNA sorting as well as siRNA specialization.** With the expanded list of

miRNAs, we next continued with the examination of the differences between NveAGO1 and NveAGO2 sRNA cargo. Independent AGO duplications occurred several times in bilaterians and a frequent outcome for such duplications is specialization in carrying either miRNAs or siRNAs (but not both) as seen in insects and nematodes[29,30]. This enables a separation of the endogenous gene regulation pathway from the antiviral one, as target silencing of these two target types requires different modulation levels and kinetics as well as different partner proteins[31].

Our initial expectation was to identify a similar type of specialization in Hexacorallia as it is known that endogenous siRNAs (endo-siRNAs) are present in *Nematostella*[9,32]. Using ShortStack[33,34] we identified endo-siRNAs in our libraries (Supplementary Data 4) and quantified their normalized read counts in NveAGO1 and NveAGO2 IP from three developmental stages (Fig. 4a–c). These results confirmed our hypothesis as NveAGO2 but not NveAGO1 is enriched with endo-siRNAs throughout development. To further confirm this hypothesis, we quantified endo-siRNAs that were annotated independently in a

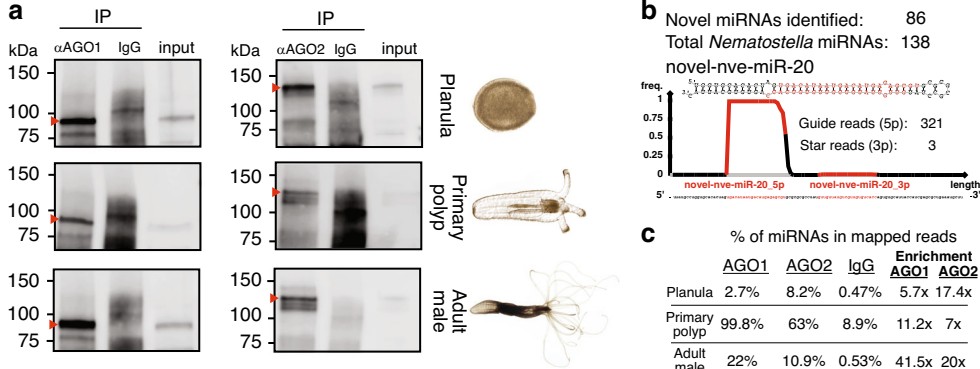

**Fig. 3 Novel miRNAs identification by AGO-IP. a** Western blot validations with αNveAGO1 and αNveAGO2 for the presence of NveAGO1(left, ~96 kDa) and NveAGO2 (right, ~122 kDa) following the IPs from three distinct developmental stages (bands corresponding to AGOs are indicated by red arrowheads). Each experiment was carried out twice independently. **b** Number of *Nematostella* miRNAs including novel miRNAs, and an example of a newly identified miRNA. **c** Average percentage of miRNA reads in sRNA libraries generated from extracts of NveAGO1, NveAGO2, and control IgG IPs. miRNA percentage was calculated from reads that mapped to the *Nematostella* genome. On the right is the enrichment in each of the NveAGO-IPs compared to control IgG.

previous study[32]. Although this list was not generated by AGO-IP sequencing, these annotated endo-siRNA were enriched in our AGO2-IP data over AGO1-IP and the negative control (Supplementary Fig. 3, Supplementary Data 4). Surprisingly, when we looked at the miRNA abundance in the two *Nematostella* AGOs, we found both to heavily load miRNAs (Fig. 4a–c, lower panels), meaning that NveAGO2, is a carrier of both miRNAs and siRNAs. Non-specialized AGOs carrying both miRNAs and siRNAs are present in at least some mammals[35,36], but this seems to be a rare phenomenon in bilaterians. Some animals possess more than one miRNA-specific AGO. In vertebrates, most of which seem to lack a widely functional siRNA pathway (but see also opposing views[37]) the specialization of each of the four miRNA-AGOs is not fully understood. They all seem to carry the same miRNAs[38], and seem to somewhat differ in their functions. For example, vertebrate AGO2, but not the others, contains an active catalytic domain that enables to promote slicing of fully complement targets[39] and to enable non-canonical maturation of some miRNAs[40–42]. In *C. elegans*, Alg5 was shown to carry some unique miRNAs that are not carried by the other two miRNA-specific AGOs of this species[43]. In this case, the specificity is enabled via tissue-specific co-expression of these particular miRNAs with Alg5 in the germline. In *Nematostella*, the two AGOs are expressed in all tissues and developmental stages[28] (Supplementary Fig. 4a, Supplementary Data 5). Nonetheless, when we tested the relative preference of individual miRNAs in NveAGO1 and NveAGO2 (calculated as explained in the "Methods" section), we were surprised to find two populations of miRNAs. The first set of miRNAs is preferentially loaded into NveAGO1, while the other is enriched in NveAGO2 (Fig. 4d–g). While the biological replicates correlated among themselves, it was clear that the occupancy of NveAGO1 and NveAGO2 miRNAs did not correlate with one another (Supplementary Fig 5); NveAGO1 miRNAs were consistently under-represented in NveAGO2 and vice versa with the exception of two miRNAs: (a) miR-9446 is enriched in NveAGO1 in adult males, and in NveAGO2 in primary polyps (b) miR-2040a is enriched in NveAGO1 in early planulae and primary polyps, and in NveAGO2 in adult males (Fig. 4d–f, Supplementary Data 4). These exceptions suggest that some developmentally regulated elements may contribute to the AGO-selectivity. We characterized the differential expression of NveAGO1 and NveAGO2 miRNAs across the three developmental stages and similarly to the previously described

*Nematostella* miRNAs[9], novel miRNAs also exhibit differential temporal expression (Supplementary Fig. 4b), further supporting the role of the miRNA pathway in *Nematostella* development. Interestingly, some miRNAs exhibited opposite strand selection in the two AGO proteins (Supplementary Fig. 6).

To elucidate the effect of AGO-depletion on miRNAs, zygotes were injected with MOs against either NveAGO1 or NveAGO2 and sRNA libraries were prepared from 3 days old planulae. Strand selection (ratio between guide fold-changes to star fold-changes) was significantly affected for AGO1 miRNAs in AGO1 KDs but not for AGO2 miRNAs in AGO2 KDs (Supplementary Fig. 7a, Supplementary Data 6). These results show that when the carrier of NveAGO1 miRNAs is depleted, miRNA-guide levels are more affected than the stars. When we examined the read counts of guide strands of miRNAs that strongly occupy each AGO (>90% preference), levels of 6 out of 7 analyzed AGO1 miRNAs were reduced in AGO1 KD, however this result is not statistically significant ($P = 0.06$) (Fig. 4h, upper panel). AGO2 miRNAs were significantly downregulated in AGO2 KD ($P = 0.013$) and significantly upregulated in AGO1 KD ($P = 0.0012$) (Fig. 4h, lower panel). Examining the levels of all miRNA guides showed a similar trend: in AGO1 KD there was no significant effect, while in AGO2 KD miRNAs levels were significantly downregulated (Supplementary Fig. 7b).

These results uncover a previously uncharacterized type of AGO subfunctionalization: a wide-scale differential miRNA sorting between the two AGOs, with NveAGO2 serving as a dual-functioning carrier of both miRNAs and endo-siRNAs.

**Characterization of NveAGO1 and NveAGO2 miRNAs**. The molecular basis for sRNA sorting into AGOs was uncovered in *Drosophila* where it was shown that central mismatches in miRNA duplexes (positions 9–11) enable their loading into the *Drosophila* miRNA-carrier: DmeAGO1, while siRNA duplexes (which usually do not contain these mismatches) are guided into the specialized siRNA carrier: DmeAGO2[44–46]. In *Arabidopsis*, mismatches at positions 9–12 facilitate miRNAs sorting[47] and additionally the identity of the first 5′ nucleotide of the guide strand plays an important role in miRNA sorting[48]. Thus, we generated NveAGO1 and NveAGO2 miRNA sequence signatures for miRNAs with >70% preference levels to a single AGO and found that unlike plants, the identity of the 5′ terminal nucleotide for both *Nematostella* miRNA populations is U (Fig. 5a). Next, we

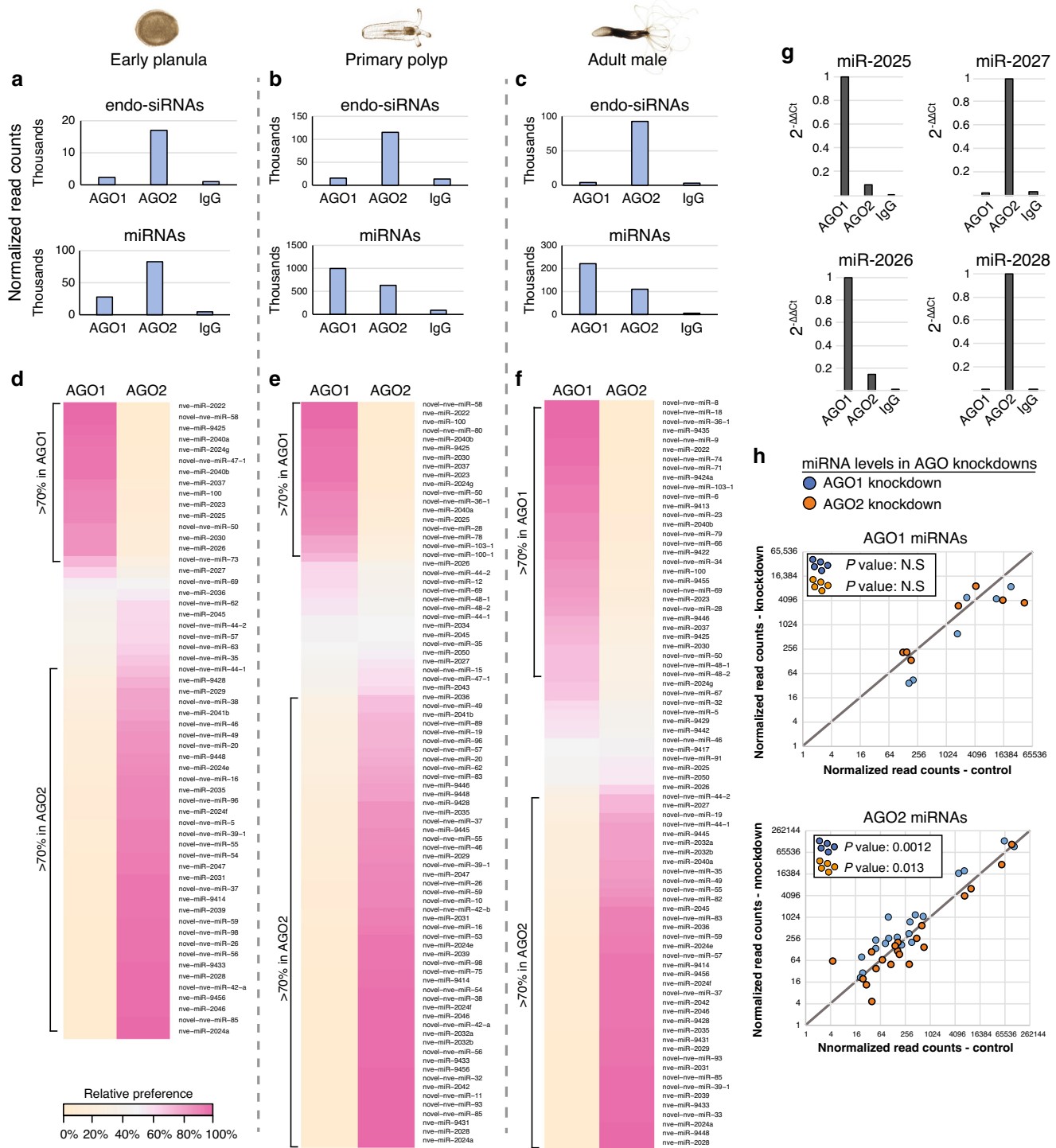

**Fig. 4 miRNA and endo-siRNA distribution between the two *Nematostella* AGOs. a–c** Normalized read-counts of miRNAs (lower panels) and endo-siRNAs (upper panels) in sRNA libraries generated from *Nematostella* AGO IPs. Each bar represents the average of two distinct biological replicates normalized as described in the "Methods" section. **d–f** Heat map representing the relative preference of individual miRNA guides to NveAGO1 or NveAGO2, validated by qPCR for 9-day-old primary polyps (**g**) for two NveAGO1 and two NveAGO2 miRNAs. **h** Levels of miRNAs with >90% preference for NveAGO1 (upper panel) and for NveAGO2 (lower panel) in AGO1 knockdowns (compared to control, blue dots) and AGO2 knockdowns (orange dots). A two-tailed binomial significance test was used to determine whether miRNA levels change following AGO knockdown (compared to their levels in controls, Supplementary data 6).

aligned the miRNAs of each NveAGO and calculated for each position of the guide-strand the frequency of mismatches between guide and star. For NveAGO1 miRNAs, there is a noticeable tendency to be enriched with mismatches in positions 10–12 compared to NveAGO2 miRNAs (Fig. 5b, Supplementary Data 4). 65% of NveAGO1 miRNAs (26/40) tend to have at least one central mismatch, compared to only 18% of NveAGO2 miRNAs (11/60). This is comparable to *Arabidopsis* where 65.7% of AthAGO1 enriched miRNAs (23/35) contain central mismatches in their duplexes compared to 20% of AthAGO2

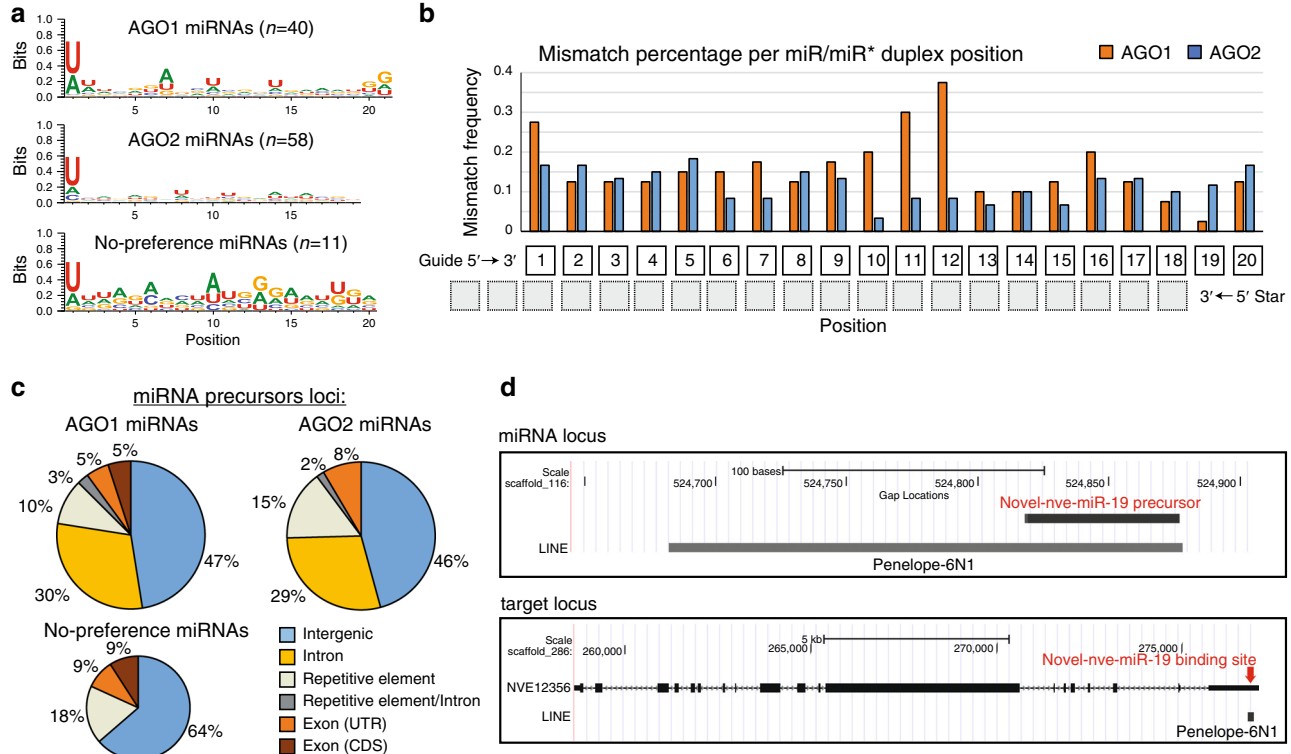

**Fig. 5 Characteristics of NveAGO1 and NveAGO2 miRNAs and the origin of some miRNA targets. a** Signature of miRNAs sequences that have >70% preference for NveAGO1, NveAGO2, or no such preference. Sequence logos were generated using WebLogo3[74]. In NveAGO2 miRNAs ≥19 nt were analyzed (58/60) **b** Mismatch frequency for miR-guide's strand positions when it forms a hybridized duplex with the star strand. **c** Genomic loci of miRNA precursors. **d** An example of a miRNA precursor positioned within a repetitive element (upper panel), and its putative target's binding site which shows the integration signature of the same repetitive element (lower panel).

miRNAs (3/15)[47]. In *Drosophila* ~80% of surveyed miRNA duplexes (80/102) exhibited central mismatches[49]. This result suggests that the molecular basis for discerning miRNAs from siRNAs in bilaterians and plants is similar to the one of *Nematostella*. However, additionally, in *Nematostella* the mechanism enables the segregation of two large sets of miRNAs. Notably, NveAGO1 miRNAs are slightly but significantly longer than NveAGO2 miRNAs (average of 22.2 and 21 nt, respectively, $P <$ 0.001 Student's $t$-test) (Supplementary Data 4). miRNAs that present no strong preference (<70%) to a specific AGO paralog exhibit an intermediate average length of 21.5 nt.

In a previous study, we discovered that similarly to plant miRNAs, and unlike bilaterian miRNAs, miRNA stabilization in *Nematostella* is mediated through the methylation of their 3′ ends by the methyltransferase HUA ENHANCER1 (HEN1). In that study, we identified two clusters that represented heavily methylated miRNAs and weakly methylated miRNAs[8]. In light of the evidence of the current study, that miRNAs are differentially distributed between the two AGOs, we decided to test whether the methylation status of AGO1 miRNAs differs from that of AGO2. To this end, we reanalyzed oxidized vs non-oxidized sRNA libraries from three developmental stages and found that the heavily methylated miRNAs consisted mostly of NveAGO1 miRNAs, while the weakly methylated miRNA cluster represented mostly miRNAs of NveAGO2 (Supplementary Fig. 8, Supplementary Data 7).

In mammals, approximately one third of the miRNA encoding loci are positioned within introns of protein-coding genes[50], while most of the rest are encoded from independent intergenic units. In plants intronic miRNA loci are rare[11]. We find that in *Nematostella*, similarly to mammals and unlike in plants, about one third of miRNA precursors are positioned within introns of

protein-coding genes, and combined with intergenic loci, comprise the vast majority of precursors (Fig. 5c). Very few precursors are positioned within the exons of protein-coding genes. Interestingly, some miRNA precursors are positioned within repetitive elements (REs). Many of these precursors (11/ 20) have predicted targets that contain signatures of the REs that host the miRNA precursor (Fig. 5d, Supplementary Data 8). A similar mechanism was previously described in mammals where some protein-coding genes can be regulated by piRNAs, via retrotransposition of REs to their 3′ UTRs[51]. Our results suggest that in *Nematostella* new miRNA targets can originate from the transposition of REs that host miRNA-precursors into the UTRs of protein-coding genes. A deeper look at predicted targets of NveAGO1 and NveAGO2 miRNAs enabled us to identify a surprising signature that might serve as a clue for understanding the evolutionary origin of miRNA precursors in Hexacorallia, as described below.

**Origin of miRNA precursors in *Hexacorallia*.** miRNA precursors are believed to emerge and evolve differently in plants and animals (bilaterians) (reviewed by Axtell et al.[11]). In plants, but not in bilaterians, the loci of some miRNA precursors exhibit homology to the target of the miRNA, which extends beyond the recognition site by the miRNA-guide strand. Hence, it was suggested that in plants new miRNA precursors are born from their own targets, when these coding genes go through inverted duplications, which upon transcription can generate hairpin precursors that can be recognized by sRNAs biogenesis machinery, and thus generate sRNA that target with high complementarity the gene from which it originated[11,52]. In bilaterians, where miRNA-target recognition is mostly restricted to a short

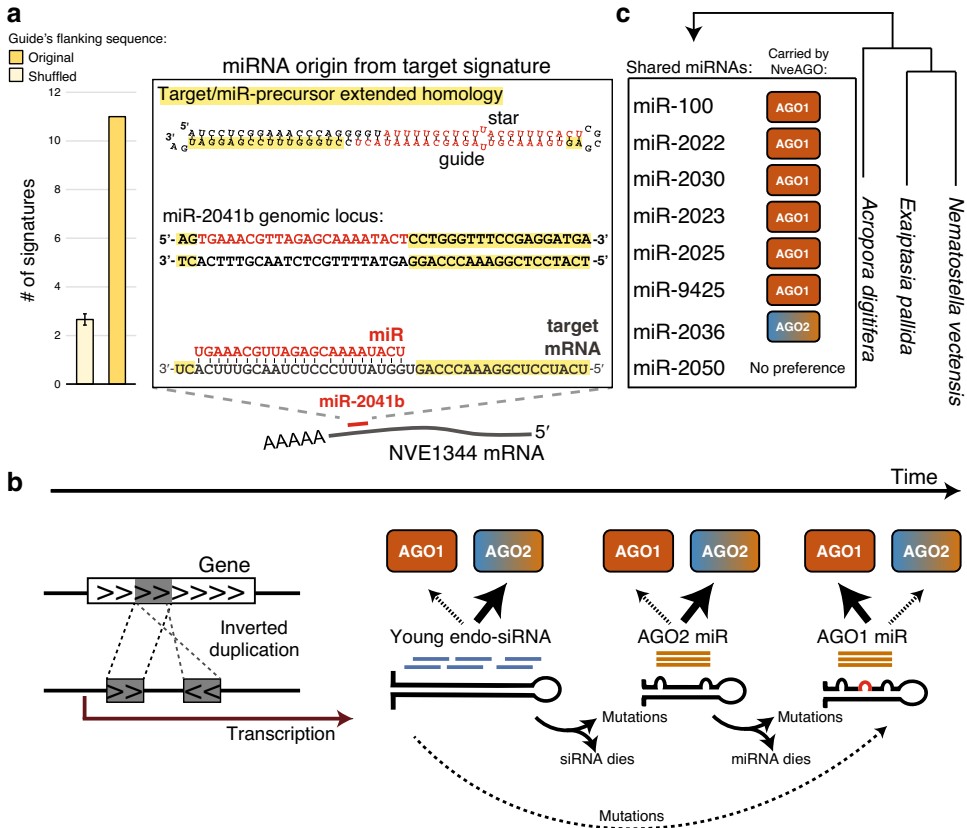

**Fig. 6 Model for origin and evolution of miRNA-precursors in *Hexacorallia*. a** Extended homology between miRNA precursors and their targets. The number of precursors with extended homology was calculated for the list of 138 miRNAs, and an average for 50 lists with shuffled sequences of the Guide's flanking regions (Data are presented as mean values +/− SE for the shuffled sequences). A signature is exemplified for miR-2041b and its predicted target NVE1344 (right panel). **b** The model suggests endo-siRNA origin (that load into hexacorallian AGO2) from inverted duplications of coding genes. Next, precursors acquire mutations that result in processing by the miRNA biogenesis machinery, and such precursors are still loaded into hexacorallian AGO2. Additional mutations such as a central mismatch in duplexes might result in the miRNA being directed to hexacorallian AGO1. **c** miRNAs shared by sea anemones and reef-building corals and their AGO preference in *Nematostella* as discovered in this study.

stretch of nucleotides at the 5′ of the miRNA, miRNAs that are highly complementary to their targets expose their unprotected 3′ end which results in the degradation of the miRNA by exonucleases[53]. In plants, where high complementary of miRNAs to their targets is frequent, the 3′ ends of miRNAs are methylated and thus protected from exonucleolytic activity[54,55].

We postulate that a plausible explanation for the difference between plant and bilaterian evolutionary origins of miRNA precursors stems from the selection against highly complementary miRNA-target interactions in bilaterians. Hence we decided to test whether in *Nematostella* (where high complementarity is frequent[9], and miRNA are stabilized by methylation[8]) miRNA precursors exhibit a signature of origin from their own targets.

To this end, we examined the list of miRNA precursors with extended homology to their predicted targets (see "Methods"). In total, 11 miRNAs exhibited such signatures: 1 AGO1 miRNA and 10 miRNAs that belong to the categories of "weakly expressed", "no AGO preference", and AGO2 miRNAs (Fig. 6a, Supplementary Data 8, 9). To estimate the levels of false-positive signatures, we generated fifty lists of *Nematostella* miRNAs where we shuffled the RNA sequences that are flanking the guide strand. On average, each list generated 2.5 false positives, never reaching a value of 11 signatures (Fig. 6a). In previous studies, Nozawa et al. showed that while in *Drosophila* none of the miRNA genes exhibit origin from targets[56], in plants ~11% (24/226) of miRNA families exhibited a signature of being born from the protein-coding genes that they target[57]. This number is comparable to our

finding in *Nematostella* (~8%, 11/138 miRNAs). Therefore, we propose that in cnidarians, similarly to plants and unlike bilaterians, miRNA precursors can originate from their own targets. This scenario implies that unlike the conservative view that animal and plant miRNA-precursors originate differently, the origin of miRNAs from targets is an ancestral mechanism that was utilized by plants and early animals, and was later lost in bilaterians. This observation piles with previous examples for commonalities between plant and cnidarian miRNAs[6,8,9,58], challenging the traditional view that the last common ancestor of plants and animals lacked a miRNA pathway.

## Discussion

Using IP, KDs, and high-throughput sequencing we revealed the developmental importance and functional differences of two hexacorallian AGOs that duplicated at least 500 MYA at the split of sea anemones and reef-building corals. This independent duplication within the sister group of Bilateria emphasizes the recurrent trend of AGO duplications and subfunctionalizations in eukaryotes, reflecting presumably its advantageous value. In most bilaterians, such duplications resulted in specialization to carry either siRNAs or miRNAs. Evolutionary analysis of AGOs from many dipteran species[30] uncovered that once such subfunctionalization is established, the miRNA-specialized AGOs do not seem to duplicate any further. On the contrary, siRNA-specialized AGOs in these species exhibit frequent duplications as

well as longer branch lengths, which might correlate with the constant need to change in the evolutionary arms race against viruses and transposons. The hexacorallian AGO duplication occurred independently of the duplications in bilaterians and yet exhibits a similar case of shorter branches for miRNA AGOs compared to that of the siRNA AGOs (Fig. 1a). Although the specialization in siRNAs of NveAGO2 was previously suggested[24], our work is the first to functionally show that NveAGO2 indeed carries this sRNA type. Unlike bilaterians, where AGO subfunctionalization was frequently accompanied by loss of catalytic domain of the miRNA AGOs, it seems that this did not occur in cnidarians, as both AGOs seem to have an intact catalytic domain. In addition, and unlike in any previous predictions, this study uncovered a dual function for NveAGO2 as we found that it loads high levels of miRNAs lacking central mismatches (Fig. 5). In *Nematostella*, the dual role of NveAGO2 is persistent throughout development (Fig. 4, Supplementary Fig. 4) and based on phylogeny (Fig. 1a) might be conserved in other sea anemones and reef-building corals.

The broad differential miRNA sorting into NveAGO1 and NveAGO2 is another unique specialization that does not typify the outcomes of AGO subfunctionalization in bilaterians. The same molecular mechanism for sorting miRNAs from siRNAs in bilaterians seems to enable the separation of one set of miRNAs from another in *Nematostella* (Figs. 4 and 5). Taking our current results together with previous findings that miRNA-mRNA high complementarity is frequent in cnidarians[9] and that unlike in bilaterians in these interactions cnidarian miRNAs remain biochemically stable[8], we suggest a new model that describes how miRNA precursors are born in Cnidaria and how they evolve in light of the presence of two subfunctionalized AGOs in Hexacorallia (Fig. 6). Our results indicate that unlike bilaterian animals, cnidarian miRNA precursors might emerge via inverted duplications of their own targets, similarly to plants (Fig. 6b) and it is plausible that this mechanism was present in the last common ancestor of cnidarians and bilaterians. In Hexacoralia, transcription of such inverted duplicated genes can generate hairpins, from which highly complementary endo-siRNAs are formed and loaded specifically into NveAGO2 (Figs. 4a–c, 6b). Next, such precursors accumulate mismatches that make them resemble miRNA precursors (e.g., 5′ homogeneity[27]), that still load into NveAGO2. Finally, mutations that facilitate NveAGO1 loading (such as central mismatches, Fig. 5b) enable these miRNAs to be preferentially loaded into NveAGO1 that does not carry endo-siRNAs.

This model suggests a long evolutionary journey until a miRNA is able to be loaded into NveAGO1. The model is further supported by the fact that the fraction of miRNA targets with extended homology to their miRNA precursor loci is higher for NveAGO2 miRNAs compared to NveAGO1 (Supplementary data 7), and finally, by the fact that the majority of the miRNAs conserved among hexacorallians are carried by NveAGO1 (Fig. 6c). While it is possible that plants and cnidarians evolved their miRNA pathways convergently, our results suggest that the scenario where the origin of miRNAs from their targets represents an ancestral phenomenon that is shared between plants and animals such as cnidarians, but was lost in bilaterians with their acquisition of seed-restricted target recognition and modulation[6], should be equally considered.

## Methods

**Animal culture and KD**. *Nematostella* polyps were grown in the dark at 18 °C in 16‰ artificial seawater and fed with freshly hatched *Artemia salina* nauplii three times a week. Induction of spawning was performed as previously described[59]. The gelatinous sack surrounding the eggs was removed using 4% L-Cysteine (Merck Milipore, USA) and followed by microinjecting the zygotes with Morpholino

antisense oligonucleotide (MO). Next, zygotes were cultured at 22 °C in 16‰ artificial seawater in the dark. The MO sequences were designed and synthesized by Gene Tools, LLC (USA).

NveAGO1 translation blocking MO1: GATTCACTGGATTATTAGAAGCCAT
NveAGO2 translation blocking MO1: TTAACAGCCTTTTGATGCTTTACGA
NveAGO1 translation blocking MO2: ACCGGTCAGTTTGTCGAGGTTATGA
NveAGO2 translation blocking MO2: CACGTCCTTTCGATTTCTTCGGCAT
Standard control MO: CCTCTTACCTCAGTTACAATTTATA

1 mM stock solution of each MO was prepared with nuclease-free water. Equal concentrations of the MOs (0.9 mM of AGO1 MO1 and AGO2 MO1, 0.45 mM of AGO1 MO2 and AGO2 MO2) of different treatments and controls were injected the same day into zygotes from the same batch to generate one biological replicate. In total, three biological replicates of ~300 injected animals were generated for each NveAGO MO and control MO. Samples for RNA or protein extraction were flash frozen in liquid nitrogen and stored at −80 °C until processed. In addition, three independent biological replicates were generated for morphological analysis and the images were collected 9 dpf.

**Antibody generation**. For NveAGO1 and NveAGO2 IP and Western blots, custom polyclonal antibodies were generated in rabbits against recombinant fragments corresponding to regions that significantly differ between the two AGOs (GenScript, USA).

NveAGO1 C-terminal region:
QVGQEQRHTYLPLEVCNIVPGQRCVKKLTDTQTSKMIRATARSAPDRERE
IRGLVKKANFDEDAYVKDFSISIGKNMVELQGRVLPPPKLVYGGKQSSPITPK
GGVWDMRGRQLFHGIEIRTWAIACFVKQQMCTEDSLRRFSNQLMKISVEQG
MPISCPPVFFRYARNPDEVERMFRRLKEAHPDLQMILVULPGKTP

NveAGO2 N-terminal:
MPKKSKGRGRGRGRGRGNPHHEKOQTLVGQQATSRNNEHKQLPKPTQTQ
QETSLSQQPYTCSSAAEAQGPLTPPNNTQGGSLTNAEPAQADTLGQKFETQ
LNLSPQSGKEQGAIPKTGARLKGNSLAPGSQGNGQFSSKNLLAQMQRTPRSA
SESQNEASSKSQQAHNNQSQAAAHQQTQAGPQQTPARSQQTPAGPQQTQ
AGPQQTPAGPQQTQAGPQQTQEGPQQTPARSQQTPAIIGSTTTADEHLAR
HRQENMEL

Specifically, each recombinant fragment was injected into three rabbits. After the third round of immunization, pre-immune and post-immune sera were sent to us for screening by Western blot against *Nematostella* lysate to identify sera specifically positive for NveAGO1 and NveAGO2 (bands of ~96 and ~122 kDa respectively). Finally, the antigens were used by the company for affinity purification from the relevant rabbits.

**AGO immunoprecipitations**. Hundred microliters of magnetic beads (SureBeads™ Protein A Magnetic Beads, Bio-Rad, USA) were washed in 1 ml of 1×PBS for five times. Five micrograms of antibodies/total IgG was added to the washed beads with 1.3 ml of 1×PBS, and left rotating overnight at 4 °C. Animals that correspond to a volume of 100 µl were frozen in liquid nitrogen and lysed (with homogenizer) using 1 ml of the following lysis buffer: 25 mM Tris-HCl (pH 7.4), 150 mM KCl, 25 mM EDTA, 0.5% NP-40, 1 mM DTT, Protease inhibitor cOmplete ULTRA tablets (Roche) and Protease Inhibitor Cocktail Set III (Merck Millipore, USA). Murine RNAse inhibitor (New England Biolabs, USA) was added whenever samples were used for downstream RNA-based applications. The DTT, Protease inhibitor and RNAse inhibitor were added fresh just before use. After 2 h rotation in 4 °C the samples were centrifuged at $16000 \times g$, 15 min, 4 °C and supernatant was collected and stored in −80 °C.

Next, the lysate was precleared as following: 100 µl of magnetic beads were washed in 1 ml of 1×PBS for three times and one time in lysis buffer and the lysate was added to the washed beads. Lysis buffer with RNAse inhibitor was added to make up 1.2 ml and the samples were incubated at 4 °C rotation for one hour. Next, the pre-cleared lysate was collected and added to the antibody-bound beads (that were pre-incubated with the antibody overnight). These samples were incubated for 2 h in rotation at 4 °C. After incubation the lysate was removed, and the beads were washed five times with the following wash buffer: 50 mM Tris-HCl (pH 7.4), 300 mM NaCl, 5 mM MgCl$_2$, 0.05% NP-40, Protease inhibitor cOmplete ULTRA tablets (Roche, Switzerland) and Protease Inhibitor Cocktail Set III, EDTA-Free (Merck Millipore). Murine RNAse inhibitor was added whenever samples were used for downstream RNA-based applications. The protease and RNAse inhibitors were added fresh just before use. Finally, for RNA extraction 1000 µl of Tri-reagent (Sigma-Aldrich) was added followed by RNA isolation according to the manufacturer's protocol with the addition of 1.5 µl of RNA-grade glycogen (Thermo Fisher Scientific, USA) when precipitating with isopropanol.

For Western blot, 40 µl of filtered double-distilled water and 20 µl of blue SDS sample buffer (New England Biolabs) were added to the beads. The samples were heated at 100 °C for 8 min and placed on ice for 1 min, then centrifuged 1 min at $23,000 \times g$ at 4 °C, and the supernatant was collected for Western blot.

**Western blot**. The samples were run on polyacrylamide gradient gel (4–15%; Bio-Rad) followed by blotting to a nitrocellulose membrane (Bio-Rad). Next, the membrane was washed with TBST buffer (20 mM Tris pH 7.6, 150 mM NaCl, 0.1% Tween 20) and blocked (5% skim milk in TBST) for 1 h at room temperature.

Purified polyclonal antibody against NveAGO1 or NveAGO2 was added to 5 ml of TBST containing 5% BSA (1:1000) (MP Biomedicals, USA), to the membrane in a sealed sterile plastic bag and incubated at 4 °C overnight. Next the membrane was washed three times with TBST and incubated for 1 h with Goat αrabbit IgG conjugated to horseradish peroxidase (Jackson's ImmunoResearch, USA) diluted to a concentration of 0.08 µg/ml in 5% skim milk in TBST. Finally, the membrane was washed three times with TBST and detection was performed with the Clarity™ ECL kit (Bio-Rad) according to the manufacturer's instructions and visualized with a CCD camera of the Odyssey Fc imaging system (Li-COR Biosciences, USA). Size determination was carried out by simultaneously running Precision Plus Protein™ Dual Color Protein Ladder (Bio-Rad) and scanning the same blot on the same system at 700 nm.

**Small-RNA sequencing**. Zygotes injected with NveAGO1, NveAGO2 and Control MOs were collected 3 dpf. Three distinct biological replicates (of ~150 animals each) were generated. Total RNA was extracted and library preparation was carried out as described in the Zamore's lab, Illumina TruSeq small-RNA Cloning Protocol April 2014 (http://www.umassmed.edu/zamore/resources/protocols/). In brief, sRNAs were ligated to 3′ and 5′ adapters that contain four random nucleotides for minimizing ligation biases. SuperScriptIII (Thermo Fisher Scientific) was used to reverse-transcribe ligated products and KAPA Real-Time Library Amplification Kit (Roche) was used for cDNA amplification. The amplified samples were run and cut from 2% low-melt agarose gels (Bio-Rad) and followed by sRNA sequencing with NextSeq500 (Illumina, USA) with read lengths of 50 nucleotides. Prior to this library preparation, four synthetic spike-ins mimicking bilaterian miRNA sequences absent from *Nematostella* genome were added to the total RNA. Two were 2′-o-methylated at their 3′ ends (mmu-miR-125a-5p and mmu-miR-148a-3p), and the other two were not methylated (cel-lin-4-5p and hsa-miR-659-5p). These four miRNAs were further used for normalization of *Nematostella* miRNA read counts in the bioinformatic analysis.

Library preparation for RNA extracted from NveAGO1 and NveAGO2 IP was size selected for 18–30 nucleotides on 15% denaturing urea polyacrylamide gel (Bio-Rad) followed by overnight RNA elution in 0.3 M NaCl. Next, library preparation was carried out using NEBNext Multiplex Small RNA Library Prep Set for Illumina kit (New England Biolabs) with modified protocol[25] for improving low-input samples and sequenced with NextSeq500 (Illumina) as described above. This procedure was carried out on two distinct biological replicates for each AGO in each of the developmental stages.

**Total RNA sequencing**. RNA was extracted with Trizol (Thermo Fisher Scientific) according to manufacturer's protocol from three biological replicates of 3 days old animals injected with control, NveAGO1 and NveAGO2 MOs. Library preparation was carried with SENSE Total RNA-seq Library Prep Kit (Lexogen, Austria) according to the manufacturer's protocol and 75 nt single-end sequencing was carried out with NextSeq500 (Illumina).

**Reverse transcription and quantitative real-time PCR (qPCR)**. Reverse transcription (RT) of miRNAs was carried out using miRCURY LNA Universal RT microRNA PCR Kit (Exiqon-Qiagen, Denmark), as instructed in miRCURY LNA RT Kit manual in three technical replicates for each miRNA. Equal amounts of RNA spike-in (Uni-Sp6) were added to the RNA and later used as an internal amplification control. RT mixture included template RNA, 5× miRCURY RT Reaction Buffer (2 µl), 10× miRCURY RT Enzyme Mix (1 µl), and nuclease-free water to make up 10 µl of total volume. The mixture was incubated at 42 °C for 1 h, then incubated at 95 °C for 5 min to inactivate the reverse transcriptase and immediately cooled to 4 °C. Real-time PCR was performed using miRCURY SYBR Green PCR Kit (Exiqon-Qiagen) according to the manufacturer's instructions with StepOnePlus Real-Time PCR System (ABI, Thermo Fisher Scientific). The qPCR mixture contained cDNA template (3 µl), 2× miRCURY SYBR Green Master Mix (5 µl), LNA primer set (1 µl) and nuclease-free water to make up 10 µl total volume. qPCR thermocycling conditions were 95 °C for 2 min, 40 cycles of 95 °C for 10 s, 56 °C for 1 min. Melt curve analysis: 60–95 °C for 15 min at a ramp-rate of 1.6 °C/s. The expression levels of miRNAs were normalized to the RNA spike-in (Uni-Sp6).

**Semi-quantitative LC-MS/MS analysis**. Sample preparation for MS analysis: After the last step of IP, the beads were washed twice with 25 mM Tris-HCl pH 8.0. The packed beads were re-suspended in 100 µl 8 M urea, 10 mM DTT, 25 mM Tris-HCl pH 8.0 and incubated for 30 min at 22 °C. Next, Iodoacetamide (55 mM) was added and beads were incubated for 30 min (22 °C, in the dark), followed by the addition of DTT (20 mM). The Urea was diluted by the addition of 6 volumes of 25 mM Tris-HCl pH 8.0. Trypsin was added (0.3 µg per sample) and the beads were incubated overnight at 37 °C with gentle agitation. The beads were spun down and the peptides were desalted on C18 home-made Stage tips[60]. Two-thirds of the eluted peptides were used for MS analysis.

nanoLC-MS/MS analysis: MS analysis was performed using a Q Exactive Plus mass spectrometer (Thermo Fisher Scientific) coupled on-line to a nanoflow UHPLC instrument, Ultimate 3000 Dionex (Thermo Fisher Scientific). Peptides were separated over a 60 min gradient run at a flow rate of 0.3 µl/min on a reverse phase 25-cm-long C18 column (75 µm ID, 2 µm, 100 Å, Thermo PepMapRSLC).

The survey scans (380–2000 *m/z*, target value 3E6 charges, maximum ion injection times 50 ms) were acquired and followed by higher-energy collisional dissociation (HCD) based fragmentation (normalized collision energy 25). A resolution of 70,000 was used for survey scans and up to 15 dynamically chosen most abundant precursor ions, with "peptide preferable" profile were fragmented (isolation window 1.6 *m/z*). The MS/MS scans were acquired at a resolution of 17,500 (target value 1E5 charges, maximum ion injection times 120 ms). Dynamic exclusion was 60 s. Data were acquired using Xcalibur software (Thermo Scientific). To avoid a carryover, the column was washed with 80% acetonitrile, 0.1% formic acid for 25 min between samples.

MS data analysis: Mass spectra data were processed using the MaxQuant computational platform, version 1.5.3.1254. Peak lists were searched against translated coding sequences of *Nematostella* gene models. The search included cysteine carbamidomethylation as a fixed modification and oxidation of methionine as variable modifications and allowed up to two miscleavages. The match-between-runs option was used. Peptides with a length of at least seven amino-acids were considered and the required FDR was set to 1% at the peptide and protein level. Protein identification required at least 2 unique or razor peptides per protein. Relative protein quantification in MaxQuant was performed using the label-free quantification (LFQ) algorithm[61]. LFQ in MaxQuant uses only common peptides for pair-wise ratio determination for each protein and calculates a median ratio to protect against outliers. It then determines all pair-wise protein ratios and requires a minimal number of two peptide ratios for a given protein ratio to be considered valid. Statistical analysis ($n = 3$) was performed using the Perseus statistical package[62]. Only sample groups with at least 2 valid values were used. Protein contaminants and proteins identified by <2 peptides were excluded from the analysis. The procedure described above was carried out on three technical replicates for each AGO-IP. MS/MS raw files, as well as results of MaxQuant analysis were deposited to the ProteomeXchange Consortium via the PRIDE55 partner repository with the dataset identifier: PXD011644.

**Bioinformatic analysis**. mirDeep2[26] core algorithm was used to identify novel miRNAs in AGO-IP samples. For quantification of miRNA counts for AGO-preference analysis, mirDeep2 quantifier.pl module was used with default parameters, and the reads were normalized to the number of reads mapping to the genome (and multiplied by 1 million) in each library. To reduce noise, only miRNAs that exceeded a threshold of minimum 20 normalized reads in at least one library were included in this analysis. The normalized read counts from two biological replicates in each AGO-IP were averaged and the relative preference levels of each miRNA in NveAGO1 and NveAGO2 were calculated by dividing the counts from each NveAGO by the miRNA's total counts in both AGOs (Fig. 4). miRNAs from the heatmap that showed no AGO-preference in any of the developmental stages were classified as "no-preference miRNAs" (Supplementary Data 4). To assess enrichment of miRNAs over non-IP libraries, the levels of NveAGO1 and NveAGO2 miRNAs from NveAGO1-IP and NveAGO2-IP (respectively) were compared to the levels of NveAGO1 and NveAGO2 miRNAs from the same three developmental stages reported by Moran et al.[9] as well the new sRNA data obtained from primary polyps (Supplementary Data 4). In NveAGO1 and NveAGO2 knockdown sRNA libraries, the quantifier.pl module was run allowing no mismatches in sRNA reads mapped to miRNA precursors. These counts were normalized to the average of the four miRNA spike-ins in each library.

For total RNA sequencing analysis, two different approaches were used: (a) filtered fastq reads were aligned to the Bowtie2[63] (version 2.3.4.1) indexed *Nematostella* genome using TopHat[64] (version 2.1.1) The number of reads mapping to each *Nematostella* gene models (https://figshare.com/articles/Nematostella_vectensis_transcriptome_and_gene_models_v2_0/807696) was extracted using featurecounts[65] 1.6.0.2. (Supplementary Fig. 2) (b) reads were aligned to the genome using STAR[66] (Fig. 2). The differential gene expression analysis was carried out using DESeq2[67].

Putative miRNA targets were predicted using psRNATarget[68] with a maximum expectation score of 2.5, allowing no gaps. miRNA origin from target signature was defined as follows: First miRNA-guide sequences were extracted including seven spanning nucleotides from the side of the stem and six from the side of the loop. Next all miRNA putative targets were aligned to the extended guide sequences using the smith-waterman algorithm with -gapopen set to 15 and -gapextend set to 2.0. Finally, we ran our homemade script that scanned all the alignment outputs and identified miRNA-target pairs with extended homology of at least six consecutive nucleotides. This script is available via GitHub (https://github.com/ArieFridrichHuji/miRNA_Origin.git). A miR with extended homology to more than one target was counted only once in this analysis. Finally, we manually inspected each positive match to validate that the miRNA and the flanking nucleotides were aligned to the correct targets site on the target's gene. In addition, we analyzed the number of signatures for miRNA guides with shuffled flanking sequences (Supplementary Data 8). The alignments are available in supplementary data 3. Prediction of miRNA target origin via retro-transposition was carried out manually by examination of miRNA loci and their putative targets.

For de novo identification of endo-siRNAs and their expression levels from IP sRNA libraries we preprocessed the raw reads to trim the adapter and then reads were filtered from known *Nematostella* miRNAs and other non-coding RNAs using Bowtie2 version 2.3.4.1. The filtered reads were subjected to ShortStack[33] (version

3.8.5) analysis with default parameters. In the final outputs, the "DicerCall" column was filtered from value: "N". This enables to eliminate sRNAs which likely are not generated via RNAi process (and frequently represent breakdown products of abundant RNAs). Finally, the MIRNA column was filtered from the value "Y", which enabled eliminating any reads that have a chance to correspond to miRNAs. For quantification of existing data of previously characterized *Nematostella* endo-siRNA[32], the lists of endo siRNA sequences were filtered from sequences that mapped to ribosomal RNAs. Next the remaining sequences were quantified in each of the AGO-IP libraries as well as in the IgG negative controls using the ShortStack count module. The reads were normalized to the reads mapped to the genome.

miRNA methylation analysis was carried as follows: sRNA libraries from our previous study[8] of three developmental stages (planula, primary polyp, and adult male) that were treated or non-treated with oxidation by periodate were quantified using the updated miRNA list presented in this manuscript with mirDeep2. NveAGO1 and NveAGO2 miRNAs with raw reads lower than 50 were removed from this analysis to reduce noise. For each of the remaining miRNAs we normalized their read counts and plotted their levels in the non-oxidized and oxidized libraries. To test if miRNA levels significantly drop after oxidation, we carried out two-tailed student's t-test for two dependent means.

**Phylogenetic analysis**. The AGO amino acid sequences were aligned using MUSCLE[69] and low certainty alignment regions were removed by TrimAl[70] using the –automatic1 for heuristic model selection. ProtTest was used to find the most suitable model for phylogeny reconstruction[71]. The maximum-likelihood (ML) phylogenetic trees were constructed using PhyML with the LG + I + G model and support values were calculated using 100 bootstrap replicates[72]. A Bayesian tree was constructed using MrBayes version 3.2.1[73] with the WAG + I + G model and the run lasted 5,000,000 generations with every 100th generation being sampled. The Bayesian analysis was estimated to reach convergence when the potential scale reduction factor reached 1.0. Sequences are available in Supplementary Data 10.

**Reporting summary**. Further information on research design is available in the Nature Research Reporting Summary linked to this article.

## Data availability

All the datasets generated during the current study are available either in the Supplementary information or in the GEO, SRA, and ProteomeXchange Consortium repositories under the identification numbers GSE144203, PRJNA658931 and PXD011644, respectively. Source data are provided with this paper.

## Code availability

The code generated in this study is available at 'GitHub https://github.com/ArieFridrichHuji/miRNA_Origin.git'.

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

## Acknowledgements

The authors are grateful to the following researchers of the core facilities of the Alexander Silberman Institute of Life Sciences, The Hebrew University: Dr. William Breuer (Interdepartmental Unit) for his help with mass spectrometry and Dr. Michal Bronstein and Ms. Adi Turjeman (The Center for Genomic Technologies) for their help with high-throughput sequencing. The authors would like to thank Prof. Liran Carmel (Department of Genetics) for his advice regarding statistical analysis, and Dr. Joachim Surm (Department of Ecology, Evolution and Behavior) for his help with bioinformatic analysis. This work was funded by the European Research Council Starting Grant (CNI-DARIAMICRORNA, 637456) to Y.M.

## Author contributions

Y.M. conceived the study; A.F., V.M., and Y.M. designed the experiments; M.L. performed the IP and small-RNA-seq experiments for adult anemones, R.A. performed the western blots and preparation of proteomic samples and A.F. and V.M. performed all other experiments and analyses; A.F. and Y.M. wrote the original draft; All authors were involved in reviewing and editing the text; Y.M. managed the project and recruited the funding.

## Competing interests

The authors declare no competing interests.
