## [Peer Review File · Nature Communications]

Reviewers' comments:

Reviewer #1 (Remarks to the Author):

This is a nicely executed and comprehensive study exploring the function of two Argonuate (Ago) proteins and their associated miRNAs in *Nematostella*. Much of the work on miRNAs has focused on bilaterian animals and plants, and these two groups have significant differences in the way miRNAs function. To better understand how the miRNA pathway has evolved, the authors have focused on *Nematostella* which is a cnidarian; this phylum is the sister group to bilaterians and therefore sits at an informative phylogenetic position. They find that *Nematostella* has two Ago proteins due to a duplication that occurred in the common ancestor of all Hexacorallia over 500 million years ago. The maintenance of both homologs implies functional importance of these two Ago proteins perhaps through sub-functionalization. Through morpholino knockdown experiments, the authors demonstrate that each Ago protein is indeed critical for development. The authors generated antibodies to recognize each Ago protein and very convincingly demonstrated their specificity. These antibodies were used to isolate and sequence small RNAs bound to each Ago protein. The authors found that Ago1 or Ago2 have a different complement of miRNAs that associate. Ago2 also interacts with siRNAs. These data support sub-functionalization of Ago1 and Ago2. By examining the sequences of these miRNAs and their targets, the authors find support for a model in which miRNAs evolve from gene duplications and are initially associated with Ago2. Subsequently, the acquisition of particular mismatches in the miRNA duplex shift the binding preference to Ago1. Overall, this is a very careful and comprehensive study that fills an important gap in knowledge. The manuscript is well-written, and the data are convincing. I have only a few comments/suggestions for improvement.

Minor issues that should be addressed:

1. Figure 1A – Based on the colors coding, it looks like Nco-Ago2 is a soft coral homolog. If that is true, does this mean that the Ago2 duplication is actually present in all Anthozoans and not limited to the Hexacorallia? Please clarify.
2. For the morpholino knockdown experiments, please provide western blot analysis of both Ago proteins so that the specificity of the morpholinos is clear.
3. The description of the model in the last paragraph is a little confusing and could use some expansion/clarification. In the common ancestor to all cnidarians Ago1 was present and at least a small number of cnidarian-specific miRNAs were present. Therefore, am I correct in understanding that the model doesn't explain the origin of the miRNA pathway in general, but rather explains how new miRNAs evolve in the Hexacorallian lineage?
4. RNA-seq experiment (Table S2): Why not perform DGE and provide a list of genes that show significant changes?

5. I found Table S5 a little bit difficult to follow. It seems the counts for the “star” reads are not included? I could only find the fold change in the ratio of guide to star reads. Perhaps I missed it? If not, it could be helpful to include the counts that were used to calculate the fold change. (Generally, the tables were very clear and easy to follow, this was my only issue)

6. Figure 3G: Mention in the legend which stage the qPCR validation was performed in.

Things that might be interesting to include/discuss (but not essential):

1. It would be of interest to check the subcellular localization of the Ago proteins if the antibodies work for immunofluorescence. Presumably the Ago proteins are cytoplasmic, but this would rule out any nuclear/epigenetic functions that I believe are found in plants.

2. I really appreciate the efforts of the authors to ensure the specificity of their antibodies. I have no doubt these antibodies are specific. I noticed when I was looking at Table S1 Mass Spec results that (especially for Ago2) there are many other genes with better hits than the Ago proteins. What is the explanation for this? Is this because MS is a somewhat noisy technique and/or have you also identified some possible interacting proteins? Anything interesting to report?

3. It is stated that, “8 out of 40 NveAGO1 miRNAs and 19 out of 60 NveAGO2 miRNAs exhibited” signatures of origin from their own targets. It could be helpful to provide an estimate of the rate of miRNA self-targeting in both plants and bilaterians to contextualize these rates. Is this rate of miRNA self-targeting much higher than what is seen in bilaterians and is it comparable to plants?

4. A previous study from the senior author (Moran et al., 2014) found that miRNAs in *Nematostella* often cleave their targets, similar to plants and unlike bilaterians. In this manuscript, the authors discuss that specialization of vertebrate AGO proteins involved loss of an active catalytic domain. It seems that both *Nematostella* Ago proteins have active catalytic domains and are able to slice targets. Is that correct? That could be an interesting point of discussion, since this form of sub-functionalization has perhaps not occurred for *Nematostella* Ago proteins.

5. Regarding the discussion that occurs from line 114 to 120, it would be helpful if it was more directly stated how many miRNAs were identified in the Moran 2014 paper.

6. Lines 141-150: This disrupts the flow of the Results section and may be better moved to the Discussion section.

7. This is a matter of personal preference, but I find some of the language a little unnecessarily dramatic when describing the gap in knowledge. I think the gap in knowledge speaks for itself. I would worry that you could turn some readers off with this language. For example:

Line 15: you could remove the word “glaring”

Line 28: you could remove the word “vastly”

Line 29 you could remove the word “Markedly”

Corrections to the text:

Line 12: Remove the comma after miRNAs

Line 54: Add a comma after “Cnidaria”

Line 196: Grammatical error, change “also enables to segregate” to “enables the segregation of”

Line 221: make “sRNA” plural

Line 228: change “complement” to complementary

Reviewer #2 (Remarks to the Author):

Fridrich et al. investigate microRNA argonaute (Ago) interactions in *Nematostella*. The work has two primary motivations, first, to test the functional relevance of miRNAs in *Nematostella* (as a model for non-bilaterian animals), and second, to explore a series of questions relating to miRNA evolution. The core experiments involve careful examinations *Nematostella* Ago miRNA payloads, and the consequences of inhibition of the two argonautes present. The work suggests that the functions of Ago1 and Ago2 in *Nematostella* are different, generates significantly more robust (and thorough) descriptions of *Nematostella* miRNAs, and establishes that Ago1 and Ago2 associate with overlapping but distinct small RNA populations. Overall, the work represents a significant step forward in our understanding of small RNAs in non-bilaterian animals, and is, in principle, suitable for publication in *Nature Communications*. There are a couple of important items that should be addressed (major concerns), together with several additional more minor issues.

Major Concerns

1. The experiment described in Fig 1f is central to claims made in the manuscript. This experiment indicates that inhibition of Ago1 and Ago2 result in changes to the transcriptome, and these changes differ in the two knockdowns. The data would be substantially more convincing if the results were shown to be robust using different morpholinos – this experiment should be performed. In addition, it might be interesting to perform double knockdowns of both Ago1 and Ago2. Finally, what does PC3 look like/how much of the variance does it explain?

2. The endo-siRNA analysis is extremely cursory. At a minimum, rigorous analysis of existing data needs to be included that shows that the reads really do correspond to endo-siRNAs (based on origin from dsRNA, etc). Moreover, the analysis could be expanded using the RNAseq data to define potential targets etc.

Minor Concerns

1. line 19 – I am not sure why the authors believe that differential loading of miRNAs between AGOs is “unexpected”. Multiple organisms display such preferences (and such work is cited within the manuscript).

2. lines 36 – it would be helpful to add a sentence (and reference) indicating that slicing activity is (almost certainly) an ancient property of AGO proteins, and ancestral to plants and animals.

3. line 143 – I do not think the clause relating to vertebrates: “most of which lack a widely functional siRNA pathway” is well supported. It is true that such a widespread pathway has not been characterized, but I do not believe it is clear nor established that a widely functional siRNA pathway does not exist in vertebrates.

4. line 241 – it is a bit of reach to say that the new data challenges the traditional view.

5. line 278 – alternatively...plants and cnidarians evolved similar modes of miRNA evolution independent from one another.

6. Is the fraction of methylated smallRNAs equivalent in the Ago1 and Ago2 smallRNA populations? It might be possible to make some inferences on this point by analyzing previous data sets that measure methylation status together with the new IP-small RNA data.

Reviewer #3 (Remarks to the Author):

The manuscript by Fridrich et al, “Unravelling the developmental and functional significance of an ancient Argonaute duplication”, reports characterizations of two Argonaute paralogs in the sea anemone, *Nematostella vectensis*. The manuscript provides characterizations of *Nematostella* small RNAs and their respective loading into the two Argonaute proteins, revealing a sub-specilazation of the two Argonautes. Authors also provide experimental evidence for the Argonautes’ roles in *Nematostella* development by assessing the effects of individual AGO1 and AGO2 knockdowns on early animal development and by reporting transcriptome changes associated with each knockdown. This manuscript fills in several gaps in our knowledge, as miRNA functional roles are understudied in non-bilaterian animals. Authors propose a hypothesis for miRNA evolution, arguing against convergent evolution. The authors finish by proposing an interesting and novel model for origin and evolution of Hexacorallia miRNA precursors. Overall, this is an exciting manuscript that sheds light on miRNA and Argonaute

evolution and will be of interest to a broad audience.

I believe addressing the following comments and suggestions will further improve this manuscript.

1. Related to Figure 2C and Methods. Enrichment of AGO IP reads over IgG control reads (ex: 5.7x and 17.4x in the planula stage) supports the fact that the RNA is unbound and washed away in the IgG control. It does not, however, help the reader assess whether the AGO IP enriched miRNA reads over input. Did the authors observe an enrichment in the miRNA populations in the AGO IP over input? It is currently unclear if the input into the IP was sequenced (it appears it was not). Presence of miRNAs in AGO IP does not equal enrichment in IP over input and could change the interpretations of AGO loading. For example, a miRNA could be present in AGO IP at same level as input or be present and de-enriched, with both scenarios arguing against AGO loading with this particular miRNA. However, this observation that can only be made if the miRNA levels are assessed in the input sample used for the IP.

2. Figure 2 panel A should include information on the % input vs IP to allow the reader to gauge the efficiency of the AGO IP.

3. Figure 2 panel B will improve with an increase in the font size within the graph to make it readable.

4. Figure 3D-F: are these “guide” strands only? If yes, that should be indicated.

5. What stage was the sRNA libraries prepared from? Line 164-165, the authors state that the libraries were generated “from planulae injected with MOs”, but it’s unclear how long after the MOs injections the animals were collected. Similarly, lines 438-439, the authors mention that the animals were collected three days post fertilization, but again, it’s unclear when the MOs injections occurred relative to sample collection.

6. Since the authors have the small RNA seq data for AGO1 and AGO2 knockdowns, why not show a larger set of AGO1-specific miRNAs, certainly more than 6 in the AGO1 panel (Figure 3H)? At the same time, I believe it may be of interest to the reader to see a profile of all miRNAs in AGO1 and AGO2 knockdowns. Perhaps as a supplemental figure? Labels for some of the miRNAs would be useful to allow the reader to correlate the miRNAs between panels D-F and H. In addition, some indication of where, say, a 2-fold difference lies would be useful—perhaps as dashed lines or a distinction of the individual dots to easily orient the reader to the differences observed.

7. The alternative strand selection in AGO1 vs AGO2 is fascinating. Could the authors comment more on this observation in light of potential specialization of miRNA loading into the two AGOs? For example, do the eleven miRNAs that exhibit this behavior have a feature distinguishing them from others? What is the status of central mismatches in these 11 miRNAs?

8. Similarly, could the authors include a comment on the reduction in miRNA guides and accumulation of

miRNA stars in AGO1 KD. Is this observation limited to the eleven miRNAs that show alternative strand selection? Do the authors believe this is due to loss of the AGO1-associated miRNA strands, resulting in a shift of guide/star ratio due to still present AGO2-loaded miRNAs? And if so, why did AGO2 KD have no such effect?

9. Could the authors provide a graphical representation of Supplementary table 7, as these data are supportive of their model on miRNA origins? A statistical comparison would further highlight their conclusion regarding a higher fraction of AGO2-miRNA targets with extended homology compared to AGO1.

10. What stage was used for the Western in Figure 1? Is there any evidence for multiple AGO2 isoforms as Figure 2 appears to have multiple (3) bands both in input and IP? Could the difference be stage specificity?

11. Do the authors see good correlation between the two biological replicates for each AGO IP? I am slightly apprehensive over the use of 2 replicates only, however, if good correlation is observed, it would alleviate this reviewer's concern.

Line 227 has a superfluous comma after "postulate". Line 267 needs a comma before "from" and line 256 needs a comma after "suggested".

Data availability:

- sRNA sequencing, GEO reviewer token: crgzuoqokpbepjqd
- Transcriptome sequencing, SRA private link for reviewers: <https://dataview.ncbi.nlm.nih.gov/object/PRJNA658931?reviewer=d03qvbvbsc6kf8qg6r3o6p66p5t>
- Proteomics data: to review the ProteomeXchange identifier PXD011644 please use PRIDE Inspector (for instructions see <https://www.ebi.ac.uk/pride/help/archive/reviewers>) and use Username: reviewer73332@ebi.ac.uk and Password: JAJHyc5q
- For accessing our script testing extended homology between miRNA and its target please use GitHub: <https://github.com/ArieFridrich/miRNA-Origin>

*** For the reviewer's convenience, we provided two files for the main text: (a) PDF of our final version and (b) a PDF with track-changes that highlight the modifications to the initial version of the manuscript. All line references in this letter correspond to the lines that appear in the final PDF.

Reviewer #1 (Remarks to the Author):

This is a nicely executed and comprehensive study exploring the function of two Argonuate (Ago) proteins and their associated miRNAs in *Nematostella*. Much of the work on miRNAs has focused on bilaterian animals and plants, and these two groups have significant differences in the way miRNAs function. To better understand how the miRNA pathway has evolved, the authors have focused on *Nematostella* which is a cnidarian; this phylum is the sister group to bilaterians and therefore sits at an informative phylogenetic position. They find that *Nematostella* has two Ago proteins due to a duplication that occurred in the common ancestor of all Hexacorallia over 500 million years ago. The maintenance of both homologs implies functional importance of these two Ago proteins perhaps through sub-functionalization. Through morpholino knockdown experiments, the authors demonstrate that each Ago protein is indeed critical for development. The authors generated antibodies to recognize each Ago protein and very convincingly demonstrated their specificity. These antibodies were used to isolate and sequence small RNAs bound to each Ago protein. The authors found that Ago1 or Ago2 have a different complement of miRNAs that associate. Ago2 also interacts with siRNAs. These data support sub-functionalization of Ago1 and Ago2. By examining the sequences of these miRNAs and their targets, the authors find support for a model in which miRNAs evolve from gene duplications and are initially associated with Ago2. Subsequently, the acquisition of particular mismatches in the miRNA duplex shift the binding preference to Ago1. Overall, this is a very careful and comprehensive study that fills an important gap in knowledge. The manuscript is well-written, and the data are convincing. I have only a few comments/suggestions for improvement.

Minor issues that should be addressed:

1. Figure 1A – Based on the colors coding, it looks like Nco-Ago2 is a soft coral homolog. If that is true, does this mean that the Ago2 duplication is actually present in all Anthozoans and not limited to the Hexacorallia? Please clarify.

Answer: The reviewer is correct regarding Hco-AGO2 which is positioned closely to AGO2 of stony corals and sea anemones. However, since Hco-AGO1 is not forming such a

topology with sea anemones and stony corals it is hard to decipher whether this duplication already occurred in the last common ancestor of all anthozoans, or alternatively, whether it represents an independent duplication typifying only octocorallians. Thus, we prefer to avoid in the manuscript over-interpretation of the phylogeny and instead focus on the chronologically minimal assumption that the AGO1-AGO2 duplication happened after the split of hexacorallians from octocorallians.

2. For the morpholino knockdown experiments, please provide western blot analysis of both Ago proteins so that the specificity of the morpholinos is clear.

Answer:

As the two AGOs are completely different in their nucleotide sequences we believe there is no reason to suspect that a morpholino against one of them will bind the other. Furthermore, they were designed that they will not bind any sequence in the *Nematostella* genome which is matching with more than 17 nt, a sequence stretch insufficient for morpholino binding. Lastly, following Reviewer #2's first comment (please see below) we added another morpholino for inhibiting the translation of each AGO transcript and by this provide additional evidence for the specificity of each morpholino.

3. The description of the model in the last paragraph is a little confusing and could use some expansion/clarification. In the common ancestor to all cnidarians Ago1 was present and at least a small number of cnidarian-specific miRNAs were present. Therefore, am I correct in understanding that the model doesn't explain the origin of the miRNA pathway in general, but rather explains how new miRNAs evolve in the Hexacorallian lineage?

Answer: We thank the reviewer for this remark. As the reviewer notes the model does not explain the origin of the miRNA pathway in general. There are two phenomena the model proposes: The first is related to origin of miRNA precursors from their own targets. This part of the model we believe can be generalized to the last common ancestor of cnidarians and bilaterians, as this part depends on high-complementarity interactions between the miRNAs and their targets and does not depend on the number of AGOs. The second part of the model is as the reviewer suggested hexacorallian specific and is based on our observation regarding the type of sRNAs found in each of the two NveAGOs. To summarize: we propose that the origin of miRNA precursors from their own targets was likely present in the cnidarian bilaterian common ancestor. An AGO duplication that occurred before the split of sea anemones and stony corals enabled subfunctionalization of NveAGO1 and NveAGO2 to carry miRNAs with central mismatches (NveAGO1), and mismatchless miRNAs and endo-siRNAs (NveAGO2). We changed the text (lines 296-308) accordingly and hope the reviewer finds our elaborated explanation to be more comprehensive.

4. RNA-seq experiment (Table S2): Why not perform DGE and provide a list of genes that show significant changes?

Answer: We have now included this data in Supplementary table 2.

5. I found Table S5 a little bit difficult to follow. It seems the counts for the "star" reads are not included? I could only find the fold change in the ratio of guide to star reads. Perhaps I missed it? If not, it could be helpful to include the counts that were used to calculate the fold change. (Generally, the tables were very clear and easy to follow, this was my only issue)

Answer: We would like to point the referee to Supplementary table 5 sheet “normalized to spike x 1M”. This sheet includes the normalized reads of both guide and stars that were used for the calculation in sheet “Strand-selection”. We corrected the main text (Line 177) to make a more precise description of this calculation. For each miRNA in each AGO KD we did the following: (1) tested the average guide’s fold-change in the knockdown (control/KD), and next divided it by the average star’s fold-change. The names in Sheet “Strand-selection” include 5p or 3p for simplicity to make it easier to know which strand corresponds to the guide of each miRNA.

6. Figure 3G: Mention in the legend which stage the qPCR validation was performed in.

Answer: We added the information to the figure legend.

Things that might be interesting to include/discuss (but not essential):

1. It would be of interest to check the subcellular localization of the Ago proteins if the antibodies work for immunofluorescence. Presumably the Ago proteins are cytoplasmic, but this would rule out any nuclear/epigenetic functions that I believe are found in plants.

Answer: This could have been a nice addition. Unfortunately, we made several attempts to immunostain NveAGOs and found these antibodies not suitable for this assay. Not every antibody is good for any assay and immunostaining sometimes heavily depends on very specific fixation conditions of the targeted tissue. As this was not considered essential by the reviewer and by us we deserted this assay after several attempts.

2. I really appreciate the efforts of the authors to ensure the specificity of their antibodies. I have no doubt these antibodies are specific. I noticed when I was looking at Table S1 Mass Spec results that (especially for Ago2) there are many other genes with better hits than the Ago proteins. What is the explanation for this? Is this because MS is a somewhat noisy technique and/or have you also identified some possible interacting proteins? Anything interesting to report?

Answer: We believe that part of the explanation to this, as the reviewer suggested, is due to the technique being noisy. In these experiments we did not treat the immunoprecipitated samples with RNase (We did not want to damage the RNA cargo in any way). Since AGOs are RNA binding proteins we believe that the additional proteins might represent some other RBPs that are not directly related to the small-RNA pathway, but rather to the RNA molecules that are potentially pulled-down with the AGOs such as fragments of messenger RNAs. Additionally, as can be seen in the IgG-IP: Antibodies have a tendency to stick non-specifically to proteins, so complete elimination of noise is impossible. Due to these limitations, we prefer to avoid making any statements regarding possible interactions of AGOs with other proteins in our LC/MS-MS experiments.

3. It is stated that, “8 out of 40 NveAGO1 miRNAs and 19 out of 60 NveAGO2 miRNAs exhibited” signatures of origin from their own targets. It could be helpful to provide an estimate of the rate of miRNA self-targeting in both plants and bilaterians to contextualize these rates. Is this rate of miRNA self-targeting much higher than what is seen in bilaterians and is it comparable to plants?

Answer: This is a very interesting point. One of the main limitations of answering this is the fact that different researchers use different criteria for annotation of miRNAs and “a list” of miRNAs for the same organism might differ based on the researcher that published it. Interestingly, to answer the reviewer’s comment there is an existing estimation of the rate for

miRNAs being born from their own targets. The main benefit of this existing estimation is the fact that the same researcher made it in both animal and plant species. This work showed that in a member of bilateria (*Drosophila*) **none** of the miRNA genes exhibited sequence similarity to protein-coding genes, suggesting that bilaterian miRNA genes are not originated from inverted duplicates of their targets¹. On the contrary, when this estimation was carried out on multiple plant species, it was found that ~11% (24/226) of miRNA families exhibited a signature of being originated from protein coding genes². We have now improved our analysis by automating the process of origin from target signature identification (please see answer to Reviewer 3 comment 9). We used more stringent criteria (6 consecutive matching nucleotides instead of 5) as it enabled a very low rate of false positives. This analysis which resulted in 11 signatures enabled us to highlight even better the difference between AGO1 and AGO2 miRNAs (only 1 AGO1 miRNA exhibited the signature) and further strengthen our hypothesis regarding the long evolutionary journey of a miRNA until it is capable of loading to NveAGO1. The fraction of identified signatures in *Nematostella* is similar to the fraction that is reported in plants. Additionally, Moran et al.³ showed that the fraction of miRNAs with nearly perfect complementarity in bilaterians is negligible compared to *Nematostella*, suggesting that it is very unlikely to find such signatures of miRNAs origin from their targets in bilaterians. This information is now incorporated to the main text in lines (261-264).

4. A previous study from the senior author (Moran et al., 2014) found that miRNAs in *Nematostella* often cleave their targets, similar to plants and unlike bilaterians. In this manuscript, the authors discuss that specialization of vertebrate AGO proteins involved loss of an active catalytic domain. It seems that both *Nematostella* Ago proteins have active catalytic domains and are able to slice targets. Is that correct? That could be an interesting point of discussion, since this form of sub-functionalization has perhaps not occurred for *Nematostella* Ago proteins.

Answer: Yes, the reviewer is correct and indeed both NveAGOs have an intact catalytic domain. We have now incorporated this point into the discussion in lines 285-288 and we appreciate this suggestion.

5. Regarding the discussion that occurs from line 114 to 120, it would be helpful if it was more directly stated how many miRNAs were identified in the Moran 2014 paper.

Answer: We thank the reviewer for their comment. The information was added and now available in line: 123.

6. Lines 141-150: This disrupts the flow of the Results section and may be better moved to the Discussion section.

Answer: We re-evaluated this point and we still believe that in the current form the story is more comprehensive and that it helps the overall flow. Unless the referee strongly opposes, we would prefer to keep this structure as it is.

7. This is a matter of personal preference, but I find some of the language a little unnecessarily dramatic when describing the gap in knowledge. I think the gap in knowledge speaks for itself. I would worry that you could turn some readers off with this language. For example:

Answer: We thank the reviewer for raising this concern and we corrected the text accordingly.

Line 15: you could remove the word “glaring”

Line 28: you could remove the word “vastly”

Line 29 you could remove the word “Markedly”

Corrections to the text:

Line 12: Remove the comma after miRNAs

Line 54: Add a comma after “Cnidaria”

Line 196: Grammatical error, change “also enables to segregate” to “enables the segregation of”

Line 221: make “sRNA” plural

Line 228: change “complement” to complementary

Answer: Thank you very much for noticing these errors. The text was corrected.

Reviewer #2 (Remarks to the Author):

Fridrich et al. investigate microRNA argonaute (Ago) interactions in *Nematostella*. The work has two primary motivations, first, to test the functional relevance of miRNAs in *Nematostella* (as a model for non-bilaterian animals), and second, to explore a series of questions relating to miRNA evolution. The core experiments involve careful examinations *Nematostella* Ago miRNA payloads, and the consequences of inhibition of the two argonautes present. The work suggests that the functions of Ago1 and Ago2 in *Nematostella* are different, generates significantly more robust (and thorough) descriptions of *Nematostella* miRNAs, and establishes that Ago1 and Ago2 associate with overlapping but distinct small RNA populations. Overall, the work represents a significant step forward in our understanding of small RNAs in non-bilaterian animals, and is, in principle, suitable for publication in *Nature Communications*. There are a couple of important items that should be addressed (major concerns), together with several additional more minor issues.

Major Concerns

1. The experiment described in Fig 1f is central to claims made in the manuscript. This experiment indicates that inhibition of Ago1 and Ago2 result in changes to the transcriptome, and these changes differ in the two knockdowns. The data would be substantially more convincing if the results were shown to be robust using different morpholinos – this experiment should be performed. In addition, it might be interesting to perform double knockdowns of both Ago1 and Ago2. Finally, what does PC3 look like/how much of the variance does it explain?

Answer: We thank the reviewer for the careful examination of our manuscript. We performed the majority of requests/suggestions and believe it helped to improve our work further. We

agree that additional morpholinos (MOs) are important to clarify the main point of this experiment: to provide additional support that AGO1 and AGO2 carry distinct functions. Therefore, we followed the reviewer's suggestion to knock them down with an additional set of translation blocking MOs against NveAGO1 and NveAGO2 MOs (AGO1 MO2 and AGO2 MO2, respectively). These MOs were injected to zygotes and we validated their efficiency using Western blot (**Supplementary figure 1b**) with our specific antibodies and we also tested the phenotype (**Supplementary figure 1a**). We concluded that the set of MOs generate developmental defects similarly to the first set and next, we explored the transcriptomic signatures of the morphants. We observed that the old and new set of AGO2 MOs generate morphants with a transcriptomic signature which differs greatly from the signature of AGO1 morphants (**Figure 2**). Although AGO2 MOs generated two distinct clusters, these were on PC2 which explained a small portion of the variance. Moreover, the overall transcriptomes which are represented in the dendrogram (**Figure 2**, lower panel), are very similar for both AGO2 MOs, and are very different from the two AGO1 MOs (which cluster closely among themselves). These results are consistent for these transcriptomes whether we used STAR to align the reads (**Figure 2**), or alternatively when we used TopHat (**Supplementary figure 2**). Additionally, following the reviewer's request we add to this response the following image to show how PC3 looks like. Similarly to PC2, it explains only a relatively small fraction of the variance, and we believe it is not very informative to include in the manuscript. The reviewer's suggestion to knockdown both AGOs simultaneously is interesting, however, we believe that such results would be too complicated to interpret as each AGO greatly affects the transcriptome, and drawing conclusions from this might not contribute to showing that each AGO carries a different role. Corrections in the main text and methods were made accordingly to explain the additions (see lines: 70, 416-420).

2. The endo-siRNA analysis is extremely cursory. At a minimum, rigorous analysis of existing data needs to be included that shows that the reads really do correspond to endo-siRNAs (based on origin from dsRNA, etc). Moreover, the analysis could be expanded using the RNAseq data to define potential targets etc.

Answer: We thank the reviewer for this comment, and have now significantly expanded the analysis by quantification of endo-siRNAs that were annotated independently from us, in a

study where endo-siRNAs were predicted as the reviewer mentioned based on RNA seq data and predictions of hairpin structures that are likely to correspond to endo-siRNA⁴. We were pleased to observe that this independently generated list of endo-siRNA showed enrichment in our AGO2-IP data over AGO1-IP and IgG in all the developmental stages (**Supplementary figure 3**). Since our de-novo endo-siRNA annotation was dependent on AGO-IP data, we believe that our list represents an extended and more comprehensive list of endo-siRNA than the lists previously published. Additionally, we expanded the main text (lines: 144-147) and methods (lines 601-611) to explain the new analysis as well as the workflow that enabled us to generate a more comprehensive list of *Nematostella* endo-siRNAs. As the reviewer suggested, we tried to expand the analysis further by examination of RNA-seq data, however it turned out to be highly complicated to interpret. The main reason for that is that siRNA counts are relatively low compared to other sRNAs (such as miRNA and piRNAs) and the potential targets of these endo-siRNAs tend to be repetitive elements that are efficiently targeted by piRNAs, which are much more abundant in cnidarians^{5,6}. Understanding the functional roles of *Nematostella* endo-siRNA would require exploration in other directions that are beyond the scope of this manuscript (such as responses to various stresses, functional examination of other siRNA potential biogenesis factors etc.). Overall, we believe this comment enabled us to improve the manuscript and are grateful for that.

Minor Concerns

1. line 19 – I am not sure why the authors believe that differential loading of miRNAs between AGOs is “unexpected”. Multiple organisms display such preferences (and such work is cited within the manuscript).

Answer: The reason we consider this “unexpected” is mentioned in the main text in lines: 157-160. While in plant miRNAs can be differentially distributed between different AGOs based on their properties, in animals differential miRNA sorting is explained mainly by tissue specific miRNA expression with a tissue specific AGO that carries it. In *Nematostella* on the contrary, while both AGOs are ubiquitously expressed in all tissues (**Supplementary figure 4**), they exhibit preference to different miRNAs, and this phenomenon does not typify the outcome of AGO duplications in bilaterian species to the best of our knowledge.

2. lines 36 – it would be helpful to add a sentence (and reference) indicating that slicing activity is (almost certainly) an ancient property of AGO proteins, and ancestral to plants and animals.

Answer: We thank the reviewer for this helpful suggestion. The sentence and references were added to line 39.

3. line 143 – I do not think the clause relating to vertebrates: “most of which lack a widely functional siRNA pathway” is well supported. It is true that such a widespread pathway has not been characterized, but I do not believe it is clear nor established that a widely functional siRNA pathway does not exist in vertebrates.

Answer: To the best of our knowledge, there is only limited evidence available for a functional siRNA pathway in vertebrates (beyond mice and rats that have a unique Dicer isoform. Please see Ref.⁷). This level of evidence remains limited despite of the fact that these models are highly investigated. We believe that it strongly indicates that while some exceptions such as mice exist, it is fair to question the existence of this functional pathway in a native state in many vertebrate lineages. Nonetheless, as this is not a main message in our paper, we toned down this sentence and also added a source to enable the reader to assess by themselves

opposing views regarding native siRNAs in vertebrates. Please see changes in the text in lines 151-152.

4. line 241 – it is a bit of reach to say that the new data challenges the traditional view.

Answer: We respectfully disagree with the reviewer on this point. We mention and cite additional publications supporting commonalities in plant and animal miRNA pathway. We have now included a new citation in this comment of our work that shows a plant miRNA biogenesis factor that plays a role in miRNA biogenesis in *Nematostella*. We believe that our statement is relatively balanced (we intentionally did not use a harsher term such as “Contradicts the traditional view”) and that the word “Challenges” enables the readers to decide for themselves what is their view on this topic given the existing literature and the citations provided in this sentence.

5. line 278 – alternatively...plants and cnidarians evolved similar modes of miRNA evolution independent from one another.

Answer: As we mentioned in the previous comment, we do not make a stand regarding the scenario, but rather suggest that alternative explanations should not be ruled out. We respect the reviewer’s view and therefore toned down this sentence and rephrased it using the reviewer’s terms in order to make what we believe is a fair representation of the possible scenarios. We thank the reviewer for this comment (see edited text in lines: 313-317).

6. Is the fraction of methylated smallRNAs equivalent in the Ago1 and Ago2 smallRNA populations? It might be possible to make some inferences on this point by analyzing previous data sets that measure methylation status together with the new IP-small RNA data.

Answer: We thank the reviewer for this very interesting comment. We tested this point and found that the fraction of methylated miRNAs is not equivalent in both AGOs. Using the updated list of miRNAs, we re-analyzed our existing data from 2018⁸ from three developmental stages. We found that AGO1 miRNAs correspond to the heavily methylated miRNAs that were identified in 2018 and that AGO2 is occupied with the cluster that was identified as the “weakly methylated” miRNAs. we incorporated this information into the main text (lines 213-221), methods (lines 612-618) and in **supplementary figure 8** and **supplementary table 6**.

Reviewer #3 (Remarks to the Author):

The manuscript by Fridrich et al, “Unravelling the developmental and functional significance of an ancient Argonaute duplication”, reports characterizations of two Argonaute paralogs in the sea anemone, *Nematostella vectensis*. The manuscript provides characterizations of *Nematostella* small RNAs and their respective loading into the two Argonaute proteins, revealing a sub-specilazation of the two Argonautes. Authors also provide experimental evidence for the Argonautes’ roles in *Nematostella* development by assessing the effects of individual AGO1 and AGO2 knockdowns on early animal development and by reporting transcriptome changes associated with each knockdown. This manuscript fills in several gaps in our knowledge, as miRNA functional roles are understudied in non-bilaterian animals. Authors propose a hypothesis for miRNA evolution, arguing against convergent evolution. The authors finish by proposing an interesting and novel model for origin and

evolution of Hexacorallia

miRNA precursors. Overall, this is an exciting manuscript that sheds light on miRNA and Argonaute evolution and will be of interest to a broad audience.

I believe addressing the following comments and suggestions will further improve this manuscript.

1. Related to Figure 2C and Methods. Enrichment of AGO IP reads over IgG control reads (ex: 5.7x and 17.4x in the planula stage) supports the fact that the RNA is unbound and washed away in the IgG control. It does not, however, help the reader assess whether the AGO IP enriched miRNA reads over input. Did the authors observe an enrichment in the miRNA populations in the AGO IP over input? It is currently unclear if the input into the IP was sequenced (it appears it was not). Presence of miRNAs in AGO IP does not equal enrichment in IP over input and could change the interpretations of AGO loading. For example, a miRNA could be present in AGO IP at same level as input or be present and de-enriched, with both scenarios arguing against AGO loading with this particular miRNA. However, this observation that can only be made if the miRNA levels are assessed in the input sample used for the IP.

Answer:

We thank the reviewer for this helpful comment. Yes, we have seen an enrichment over the input and we now incorporated the data and the description in the main text (lines: 115-119, 573-576) and in supplementary table 3, sheet "IP_vs_Input". Initially, we carried out the analysis only on the primary polyp stage, where we incorporated inputs to the sequencing libraries and observed a strong enrichment (107.8 fold over input for AGO1 and 85 fold for AGO2) as we now show to the reader. When our investigation was expanded for testing earlier and later developmental stages we did not include inputs in our sRNA sequencing, as in our opinion IgG was the most appropriate approach. However, such libraries that represent inputs from all three developmental stages of *Nematostella* presented in this work have been previously made³. We reanalyzed these libraries with the same parameters that were used in the current work, and with our updated list of miRNAs. Since NveAGO1 and NveAGO2 preferentially load different miRNAs, we reanalyzed the data in the following way:

miRNAs were normalized to the total reads mapped to the genome. Next, in each developmental stage we tested enrichment of AGO1 miRNAs in AGO1 IP compared to the AGO1 miRNAs in the 2014 "input". Similarly, we tested enrichment of AGO2 miRNAs in AGO2 IP compared to AGO2 miRNAs in the 2014 "input".

The average enrichment was higher than two-fold in all combinations except early planula AGO1 miRNAs in AGO1 IP (compared to AGO1 miRNAs in Moran et al.³ inputs). Early planula stage is the developmental stage with least expression of miRNAs. Since AGO2 IP in this stage showed enrichment over the input, and both AGO1 and AGO2 IP from this stage showed enrichment over IgG, and additional stringent criteria for miRNA annotation were considered (5' homogeneity, an obvious strand selection, guide star ratio greater than 2 fold, and additional criteria that are mentioned in the main text) we believe that this sample is also adequate for performing the analysis we carried on the rest of the samples. The information is now available in **Supplementary table 3** sheet: "IP_vs_input". We would like to emphasize that we still see the IgG libraries as a more appropriate control as some antibodies and even

beads do have some intrinsic “stickiness” that can cause non-specific precipitation of protein-RNA complexes, and input libraries are not accounting for this important property.

2. Figure 2 panel A should include information on the % input vs IP to allow the reader to gauge the efficiency of the AGO IP.

Answer: We thank the reviewer for this suggestion. As mentioned in the previous response the information regarding the enrichments over inputs are mentioned in the main text as well as in supplementary table 3 sheet “IP vs Input”. Since at the stage of figure 2 presentation the readers are still not aware of the fact that each AGO is occupied with different miRNAs, we found it a bit confusing to incorporate the information of enrichments over the inputs at this stage (as the analysis over inputs test for enrichment of AGO1 miRNAs in AGO1-IP over AGO1 miRNAs in Input. Similarly, the analysis was carried out for AGO2 miRNAs in AGO2-IP over AGO2 miRNAs in the input). As we believe this can be confusing for the reader, we would strongly prefer at this stage to direct the reader in the main text and this information is now mentioned in lines 115-119.

3. Figure 2 panel B will improve with an increase in the font size within the graph to make it readable.

Answer: The font was increased according to the reviewer’s suggestion (Figures shifted see Figure 3).

4. Figure 3D-F: are these “guide” strands only? If yes, that should be indicated.

Answer: We thank the reviewer for noticing that. We now mention it in the figure legend (Figure shifted, see Figure 4).

5. What stage was the sRNA libraries prepared from? Line 164-165, the authors state that the libraries were generated “from planulae injected with MOs”, but it’s unclear how long after the MOs injections the animals were collected. Similarly, lines 438-439, the authors mention that the animals were collected three days post fertilization, but again, it’s unclear when the MOs injections occurred relative to sample collection.

Answer: We now corrected the text and made it clear that MOs were injected to zygotes which were further raised for 3 days. On day 3 we extracted sRNAs. We thank the reviewer for noticing this. See lines 175 and 489 for the corrections we made.

6. Since the authors have the small RNA seq data for AGO1 and AGO2 knockdowns, why not show a larger set of AGO1-specific miRNAs, certainly more than 6 in the AGO1 panel (Figure 3H)? At the same time, I believe it may be of interest to the reader to see a profile of all miRNAs in AGO1 and AGO2 knockdowns. Perhaps as a supplemental figure? Labels for some of the miRNAs would be useful to allow the reader to correlate the miRNAs between panels D-F and H. In addition, some indication of where, say, a 2-fold difference lies would be useful—perhaps as dashed lines or a distinction of the individual dots to easily orient the reader to the differences observed.

Answer: We thank the reviewer for this comment. Since AGO preference of 70% for a particular miRNA means ~2.3 fold difference, it still enables a non-negligible fraction of this miRNA to occupy the opposite AGO. Hence, we decided to focus only on miRNAs with a very strong preference of 90% (as mentioned in the text in line 181). We agree that adding a

representation of all miRNAs in AGO1 and AGO2 KD could be useful as a supplementary figure and have now included it (**Supplementary figure 7b**, main text line 184). This analysis resulted in a similar conclusion: While the change in AGO1 KD was insignificant, AGO2 KD resulted in a significant reduction of miRNA levels.

We believe that a potential explanation for that might stem from the different biochemical properties of the two AGOs: in AGO2 KD, AGO1 is probably able to continue sorting centrally mismatched miR/miR-star duplexes, while the levels of the rest of miRNAs (which are mostly AGO2 miRNAs) go down as AGO1 is not able to load and protect them. AGO2 on the contrary, seems to be able loading under control conditions non-mismatched duplexes but this might be not due to preference for such duplexes but simply because of losing the competition for mismatched duplexes to AGO1. In such a scenario, In AGO1 KD, highly expressed, centrally-mismatched miRNAs are still in the system and nothing prevents them from being loaded into AGO2, which results in a non-significant response of the miRNA population in this KD. However, since this suggestion is highly hypothetical, we decided to act with caution and not elaborate on it in the main text.

Labelling miRNAs comes out fine for AGO1 miRNAs panel but not for AGO2 miRNAs panel as it is much more dense with data points. Therefore we prefer to leave the figure as it is.

7. The alternative strand selection in AGO1 vs AGO2 is fascinating. Could the authors comment more on this observation in light of potential specialization of miRNA loading into the two AGOs? For example, do the eleven miRNAs that exhibit this behavior have a feature distinguishing them from others? What is the status of central mismatches in these 11 miRNAs?

Answer: We appreciate the reviewer's interest in this point which we also found to be very interesting. Unfortunately, we could not identify any significant difference for these miRNAs compared to others. Since the 5'-end hybridization is known to be responsible for the strand selection process we tried to see whether there was a unifying feature for strands that are chosen in an opposite way. However, we could not identify such a unifying feature.

8. Similarly, could the authors include a comment on the reduction in miRNA guides and accumulation of miRNA stars in AGO1 KD. Is this observation limited to the eleven miRNAs that show alternative strand selection? Do the authors believe this is due to loss of the AGO1-associated miRNA strands, resulting in a shift of guide/star ratio due to still present AGO2-loaded miRNAs? And if so, why did AGO2 KD have no such effect?

Answer: Following the reviewer's comment we found our original description of the analysis not very clear and have now corrected it to precisely explain what we have done (See lines 176-180). This analysis does not show reduction of guides and accumulation of stars, but rather the following point:

For miRNAs of each AGO, it shows the ratio of average fold-change of the guide to average fold-change of the star strand in knockdown of either AGO1 or AGO2.

Since star levels in *Nematostella* tend to be very low as strand selection is very efficient in cnidarians, this analysis tells if on average guide levels change more than star levels in the knockdowns.

For AGO1 miRNAs in AGO1 knockdown, guides of 9 out of 9 miRNAs were more affected than stars. In AGO2 knockdown, only 1 AGO1 miRNA showed this trend. This result is statistically significant, and we hypothesize that there are several factors that could contribute

to this: (a) in the absence of AGO1, these miRNAs could still be loaded to AGO2, where strand selection could occur slightly differently than in AGO1. (b) Some of these miRNAs might be competing with other abundant miRNAs that have no central mismatches and are loaded into AGO2.

Our hypothesis is that: AGO2 loads miRNA duplexes without mismatches not because it has a preference for these duplexes but rather that it loses the competition for centrally mismatched duplexes to AGO1 which has a strong preference for these duplexes. To summarize our hypothesis regarding AGO1 miRNAs, we assume that without AGO1, abundant AGO1 miRNAs might still be loaded into AGO2 where strand selection might occur differently. In addition, abundant miRNAs would still be loaded into AGO2 but less abundant miRNAs that go to AGO1 due to their central mismatches would now be less represented as there are many AGO2-miRNAs that occupy AGO2.

Regarding AGO2 miRNAs in AGO2 knockdowns, for 9 out of 14 miRNAs guides were more affected than stars. In AGO1 knockdown, only 6 of 14 miRNAs showed this trend. This result is not statistically significant but we believe it might still point towards a similar trend as discussed above for AGO1 miRNAs. It does seem that more AGO2 miRNAs are affected in this analysis in AGO2 knockdown than in AGO1 knockdown, but the lower numbers make the result insignificant. If we are correct regarding this, we would suggest that point (a) (potential differences in AGO strand selection biochemical properties) and point (b) (competition of miRNA on loading into the remaining AGO) contribute to the result showing that more AGO2 miRNAs are affected in AGO2 knockdown than in AGO1 knockdown, in spite of not being statistically significant.

We believe that while being interesting, our explanation is quite hypothetical and hence we prefer to exclude it from the text.

9. Could the authors provide a graphical representation of Supplementary table 7, as these data are supportive of their model on miRNA origins? A statistical comparison would further highlight their conclusion regarding a higher fraction of AGO2-miRNA targets with extended homology compared to AGO1.

Answer: We thank the referee for this comment. We now provided graphical representation as well as a calculation of estimated false positives (**Figure 6a, Supplementary table 7 and Supplementary file 3**). To test this we generated a script that aligns each miRNA (with its flanking sequences) to its predicted targets and tests whether extended homology exists (GitHub: <https://github.com/ArieFridrich/miRNA-Origin>). To estimate the number of false positives we generated 50 lists of miRNAs with shuffled flanking sequences (described in the text in lines: 256-264, 397, 591-595). Initially we ran the script with parameters that mimicked our manual analysis in the initial version of this paper (searching for 5 consecutive homologous nucleotides in the flanking regions). We got a similar number of miRNAs exhibiting an origin from target signature, However, we discovered that five consecutive homologous nucleotides generated high frequency of false positives (approximately 50%). Hence, we re-run our script with more stringent parameters requiring homology of 6 consecutive nucleotides. The resulted amount of signature with these criteria is 11, with a 4-fold enrichment over the list of false positives. These stringent parameters enabled us to highlight even better the difference in signatures for AGO1-mRNAs compared to the rest of the miRNAs (only 1 AGO1 miRNA exhibited the signature). Hence it further strengthens our hypothesis regarding the long evolutionary journey of a miRNA until it is capable of being loaded to NveAGO1, and supporting that miRNAs that originated from their targets are the ones that are not found in

AGO1. We are grateful for the reviewer's comment which enabled us to improve the testing of this hypothesis.

10. What stage was used for the Western in Figure 1? Is there any evidence for multiple AGO2 isoforms as Figure 2 appears to have multiple (3) bands both in input and IP? Could the difference be stage specificity?

Answer: In Figure 1 the western was carried out on 3 days old planulae. Although we are not aware of the existence of isoforms and found no support for this in RNA-seq, we cannot fully reject this possibility. Another possibility might be that the AGOs can carry RNAs of different mass, in different stages. We have now incorporated the information about the stage used in the western blot in Figure 1.

11. Do the authors see good correlation between the two biological replicates for each AGO IP? I am slightly apprehensive over the use of 2 replicates only, however, if good correlation is observed, it would alleviate this reviewer's concern.

Answer: We thank the reviewer for this helpful remark. Yes, the biological replicates clearly correlate. We now incorporated this information into the main text (Lines 163-165) and **Supplementary figure 5**.

Line 227 has a superfluous comma after "postulate". Line 267 needs a comma before "from" and line 256 needs a comma after "suggested".

Answer: We thank the reviewer for noticing that. We corrected these sentences accordingly.

References

- 1 Nozawa, M., Miura, S. & Nei, M. Origins and evolution of microRNA genes in Drosophila species. *Genome Biol Evol* **2**, 180-189, doi:10.1093/gbe/evq009 (2010).
- 2 Nozawa, M., Miura, S. & Nei, M. Origins and evolution of microRNA genes in plant species. *Genome Biol Evol* **4**, 230-239, doi:10.1093/gbe/evs002 (2012).
- 3 Moran, Y. *et al.* Cnidarian microRNAs frequently regulate targets by cleavage. *Genome research* **24**, 651-663, doi:10.1101/gr.162503.113 (2014).
- 4 Calcino, A. D., Fernandez-Valverde, S. L., Taft, R. J. & Degnan, B. M. Diverse RNA interference strategies in early-branching metazoans. *BMC Evol Biol* **18**, 160, doi:10.1186/s12862-018-1274-2 (2018).
- 5 Juliano, C. E. *et al.* PIWI proteins and PIWI-interacting RNAs function in Hydra somatic stem cells. *Proc Natl Acad Sci U S A* **111**, 337-342, doi:10.1073/pnas.1320965111 (2014).
- 6 Praher, D. *et al.* Characterization of the piRNA pathway during development of the sea anemone *Nematostella vectensis*. *RNA Biol* **14**, 1727-1741, doi:10.1080/15476286.2017.1349048 (2017).
- 7 Flemr, M. *et al.* A retrotransposon-driven dicer isoform directs endogenous small interfering RNA production in mouse oocytes. *Cell* **155**, 807-816, doi:10.1016/j.cell.2013.10.001 (2013).
- 8 Modepalli, V., Fridrich, A., Agron, M. & Moran, Y. The methyltransferase HEN1 is required in *Nematostella vectensis* for microRNA and piRNA stability as well as larval metamorphosis. *PLoS Genet* **14**, e1007590, doi:10.1371/journal.pgen.1007590 (2018).

REVIEWERS' COMMENTS

Reviewer #1 (Remarks to the Author):

The authors have provided satisfactory responses to my previous concerns, which were minor. I am in full support of publication and do not need to see this manuscript again.

Two small things I noticed that the authors should address:

Figure 1 legend: there is no explanation of the abbreviation “Hma”

Lines 267-269: I’m a little confused here. The text says there are 11 miRNAs with extended homology to predicted targets (only 10 are then described in the same sentence) – but when I look at supplementary table 7, 19 miRNAs are shown with extended homology.

Reviewer #2 (Remarks to the Author):

The authors have responded effectively to points raised during the initial round of review. The new data significantly improves the work, which was already solid. I recommend that the manuscript be accepted, and am confident that this work will be well received.

Reviewer #3 (Remarks to the Author):

The authors have adequately addressed the concerns raised by this reviewer, with one minor follow-up: it appears that the new analysis of IP-enriched vs. “input” microRNAs was done using previously published, presumably independent sample data for “input” and not, in fact, the paired input samples from which IPs were performed. While I appreciate the authors' effort to generate a comparison of their IP libraries to total RNA (non-IP) libraries to assess miRNA enrichment in the two AGO proteins, such comparison of independent samples is not valid, and the authors should not refer to them as “input” (ex: line 118). I urge the authors to more clearly state that these are independent samples.

Additional minor comment: I assume that the provided GitHub link will become populated with the script should the manuscript be accepted for publication.

Overall, this work remains an important contribution towards our understanding of functional Argonaute sub-specialization as well as microRNA evolution and fills in several previously existing gaps. This reviewer believes that the authors' responses to all the reviewers' comments improve the manuscript and bolster the stated conclusions.

Reviewer #1 (Remarks to the Author):

The authors have provided satisfactory responses to my previous concerns, which were minor. I am in full support of publication and do not need to see this manuscript again.

Two small things I noticed that the authors should address:

Figure 1 legend: there is no explanation of the abbreviation “Hma”.

- **Answer: We thank the referee for noticing this. Hma stands for *Hydra magnipapillata* and this is now indicated in the Figure legend.**

Lines 267-269: I'm a little confused here. The text says there are 11 miRNAs with extended homology to predicted targets (only 10 are then described in the same sentence) – but when I look at supplementary table 7, 19 miRNAs are shown with extended homology.

- **Answer: Regarding the point that 11 miRNAs exhibit extended homology but only 10 are discussed: We thank the reviewer for noticing this, it should have been 1 plus 10 (and not 1 plus 9). We corrected this typo.**
- **Regarding the point that 11 miRNAs exhibit extended homology, but there are 19 miRNA in the supplementary data file: In the supplementary data file there are 11 miRNAs that show extended homology, however, since few miRNAs have extended homology to more than one target, we show all their matches as well, which results in 19 total alignments in the file. We have now added a clarification in the methods that a miRNA with extended homology to more than one target was counted once:**

“Finally, we ran our homemade script that scanned all the alignment outputs and identified miRNA-target pairs with extended homology of at least six consecutive nucleotides. This script is available via GitHub (https://github.com/ArieFridrichHuji/miRNA_Origin). **A miR with extended homology to more than one target was counted only once in this analysis. Finally, we....”**

We thank the reviewer again for helping us to improve the manuscript.

Reviewer #2 (Remarks to the Author):

The authors have responded effectively to points raised during the initial round of review. The new data significantly improves the work, which was already solid. I recommend that the manuscript be accepted, and am confident that this work will be well received.

- **Answer: We thank the reviewer again for helping us to improve the manuscript.**

Reviewer #3 (Remarks to the Author):

The authors have adequately addressed the concerns raised by this reviewer, with one minor follow-up: it appears that the new analysis of IP-enriched vs. “input” microRNAs was done using previously published, presumably independent sample data for “input” and not, in fact, the paired input samples from which IPs were performed. While I appreciate the authors' effort to generate a

comparison of their IP libraries to total RNA (non-IP) libraries to assess miRNA enrichment in the two AGO proteins, such comparison of independent samples is not valid, and the authors should not refer to them as “input” (ex: line 118). I urge the authors to more clearly state that these are independent samples.

- **Answer: We thank the reviewer for their comments and have made the changes accordingly. In line 118 we moved the brackets that state “input” next to the phrase where we describe the current study (this input is dependent and was coupled to the experiment of IP from primary polypos). Next, in line 119 we clearly state that the additional data we used was independent. Finally, we corrected supplementary data 4 to clearly state which samples are inputs and which samples are independent sRNA seq data.**

Additional minor comment: I assume that the provided GitHub link will become populated with the script should the manuscript be accepted for publication.

- **Answer: The GitHub link was corrected and now works:
https://github.com/ArieFridrichHuji/miRNA_Origin.git.**

Overall, this work remains an important contribution towards our understanding of functional Argonaute sub-specialization as well as microRNA evolution and fills in several previously existing gaps. This reviewer believes that the authors' responses to all the reviewers' comments improve the manuscript and bolster the stated conclusions.

- **We were happy to learn that the reviewer found our revised manuscript valuable and thank them for helping us to improve it.**